# Earth system model simulations show different feedback strengths of the terrestrial carbon cycle under glacial and interglacial conditions

Markus Adloff[1,2,3], Christian H. Reick[1], and Martin Claussen[1,3]

[1]Max Planck Institute for Meteorology, Bundesstraße 53, 20146 Hamburg, Germany
[2]now: School of Geographical Sciences, University of Bristol, University Road, BS8 1SS, United Kingdom
[3]Meteorological Institute, Centrum für Erdsystemforschung und Nachhaltigkeit (CEN), Universität Hamburg, Germany

*Correspondence to:* Markus Adloff (markus.adloff@bristol.ac.uk)

**Abstract.** In simulations with the MPI Earth System Model we study the feedback between the terrestrial carbon cycle and atmospheric $CO_2$ concentrations under ice age and interglacial conditions. We find different sensitivities of terrestrial carbon storage to rising $CO_2$ concentrations in the two settings. This result is obtained by comparing the transient response of the terrestrial carbon cycle to a fast and strong atmospheric $CO_2$ concentration increase (roughly 900 ppm) in $C^4$MIP type simulations starting from climates representing the last glacial maximum (LGM) and pre-industrial times (PI). In this setup we disentangle terrestrial contributions to the feedback from the carbon-concentration effect, acting biogeochemically via enhanced photosynthetic productivity when $CO_2$ concentrations increase, and the carbon-climate effect, which affects the carbon cycle via greenhouse warming. We find that the carbon-concentration effect is larger under LGM than PI conditions because photosynthetic productivity is more sensitive when starting from the lower, glacial $CO_2$ concentration and $CO_2$ fertilization saturates later. This leads to a larger productivity increase in the LGM experiment. Concerning the carbon-climate effect, it is the PI experiment in which land carbon responds more sensitively to the warming under rising $CO_2$ because at the already initially higher temperatures tropical plant productivity deteriorates more strongly and extra-tropical carbon is respired more effectively. Consequently, land carbon losses increase faster in the PI than in the LGM case. Separating the carbon-climate and carbon-concentration effects, we find that they are almost additive for our model set-up, i.e. their synergy is small in the global sum of carbon changes. Together, the two effects result in an overall strength of the terrestrial carbon cycle feedback that is almost twice as large in the LGM experiment as in the PI experiment. For PI, ocean and land contributions to the total feedback are of similar size, while in the LGM case the terrestrial feedback is dominant.

## 1 Introduction

At the last glacial maximum (21 000 yrs before present, from now on LGM), global mean surface temperature was 4 to 5°C lower than today (Annan and Hargreaves, 2013). Vegetation was not only less widespread but also primary productivity was smaller (Prentice and Harrison, 2009). This was the consequence of the lower $CO_2$ concentrations during those times (about 200 ppm less than today), acting physically via the resulting lower temperatures (greenhouse effect), and biogeochemically

via the reduced photosynthetic activity due to less available $CO_2$ in the atmosphere (reduced $CO_2$ fertilization) (Prentice and Harrison, 2009). From measuring isotopic carbon composition in ocean sediment cores (Bird et al., 1996) and the isotopic oxygen composition of air trapped in ice cores (Ciais et al., 2012) it has been estimated that terrestrial carbon storage was several hundred gigatons less than today. This is consistent with less primary productivity whose effect on carbon storage must have been larger than the reduction in soil respiration by the lower temperatures (Prentice and Harrison, 2009). This describes how $CO_2$ shaped the terrestrial carbon cycle at the LGM. But the terrestral carbon cycle acts also back on the atmospheric $CO_2$ concentration. Hence one may wonder whether the strength of this feedback was different from today at glacial times. This is what we investigate in the present paper by performing Earth system simulations for conditions of the last glacial maximum and pre-industrial (PI) times. Indeed one could ask this question also for the oceanic carbon cycle component, but this paper focuses on the terrestrial component, which will be shown to dominate the difference in feedback strength between the two Earth system states.

To quantify the feedback between carbon cycle and climate, Friedlingstein et al. (2003) introduced two sensitivities characterizing the change in stored carbon (terrestrial and/or oceanic) due to different drivers: due to biogeochemical effects of changed atmospheric $CO_2$ concentration, called the *carbon-concentration effect* measured by the $\beta$ sensitivity [PgC/ppm], and due to climate change, called the *carbon-climate effect* measured by the $\gamma$ sensitivity [PgC/K]. For recent climate, these sensitivities have been quantified in numerous Earth system simulations, especially within the international Coupled Climate Carbon Cycle Model Intercomparison Project ($C^4MIP$) (see e.g. Friedlingstein et al. (2006); Ciais et al. (2013)). Attempts to quantify carbon cycle sensitivities for perturbations of climates from even earlier times are rare. The few observational studies relate reconstructions of atmospheric $CO_2$ concentrations to reconstructions of temperature (see Friedlingstein (2015) for a review), but the resulting 'observed' sensitivity estimates of atmospheric $CO_2$ concentration to temperature typically involve the combined carbon-concentration and carbon-climate effect and are thus neither measuring $\beta$ nor $\gamma$ as defined by Friedlingstein et al. (2003). An exception is the study by Frank et al. (2010), who considered temperature and $CO_2$ reconstructions for the last Millennium before the industrial revolution: Their estimate should be a good proxy for $\gamma$ since during this period the changes in atmospheric $CO_2$ concentration have been only a few ppm so that the carbon-concentration effect should be negligible. The resulting $\gamma$ sensitivity turns out to vary in time showing values compatible with the low end of the range of values found in the $C^4MIP$ studies for recent climate. Jungclaus et al. (2010) obtained similar values for $\gamma$ from Earth system simulations of the last Millennium. The compatibility of those $\gamma$ values for the last Millennium with those from the $C^4MIP$ for recent climate may not be that surprising since the climates differ only moderately. On the other hand, the $C^4MIP$ values result from simulations that perturb the PI climate dramatically ($\approx$ quadrupling of atmospheric $CO_2$ concentration), while those for the last Millenium are obtained from historical climate and $CO_2$ variations (observed Frank et al. (2010) or simulated Jungclaus et al. (2010)) that are rather moderate so that it is unclear what such a comparison of $\gamma$ values actually means. To assure comparability, in the present study we adopt the $C^4MIP$ methodology to determine carbon cycle sensitivities for past *and* recent times.

While there have been attempts to determine climate sensitivity for various climates of the deep past (see e.g. PALEOSENSE (2012)), similar studies for carbon sensitivities are apparently missing. Nevertheless, for the climate during the LGM studied here, the underlying carbon-concentration and carbon-climate effects have been isolated in simulations to understand their

separate importance for shaping the geographical distribution of vegetation as compared to today (e.g. Claussen et al. (2013); Woillez et al. (2011)). While in these studies it was sufficient to simulate time slices for past and recent times, transient simulations are needed to determine carbon cycle sensitivities that could be compared to $C^4MIP$ values. In the present study we employ a fully coupled General Circulation Model including dynamic vegetation for transient simulations starting either from a climate state representing the LGM or from PI conditions and forced by a strong increase in atmospheric $CO_2$. Letting the $CO_2$ act either physically or biogeochemically, we isolate the individual contributions from the carbon-concentration and carbon-climate effects to changes of the terrestrial carbon budgets. Using this $C^4MIP$ type experiment design we quantify their contribution not only by computing $\beta$ and $\gamma$ for land carbon, but also by performing a factor analysis following Stein and Alpert (1993) to investigate in particular the additivity of the two effects which is a pre-condition to obtain from those two sensitivities the feedback strength.

The paper is organized as follows: First we lay out the design of our simulation experiments. Next, in section 3, we describe the mathematical framework used for our factor and feedback analysis. The analysis of the simulation results starts in section 4 with a description of the two initial climate states representing the LGM and PI conditions (1850 AD). This prepares for the analysis of the transient simulation in section 5, that contains the main results of our investigation. By applying the factor and feedback analysis we demonstrate that the intensity of the considered feedback is very different for last glacial maximum and recent climate and identify the underlying mechanisms explaining the observed differences in system behaviour. The paper concludes with a critical discussion of our results.

## 2 Experiment set up

To quantify the feedback between carbon cycle and atmospheric $CO_2$ concentrations we combine the $C^4MIP$ experiment design (Ciais et al., 2013, Box 6.4) in the variant of concentration driven simulations with a factor separation following Stein and Alpert (1993). Technically, we proceed by investigating the reaction in climate and carbon cycle to a prescribed strong rise in atmospheric CO2. More precisely, we perform a set of four simulations called "*ctrl*", "*clim*", "*conc*", and "*full*". While for the quantification of the feedbacks by the $C^4MIP$ approach only three of these simulations are needed, by using the full set of all four simulations we are able to demonstrate that – in contrast to other models (Gregory et al., 2009; Zickfeld et al., 2011; Schwinger et al., 2014) – the linearity assumption implicit to the $C^4MIP$ feedback analysis is indeed justified for our model. Starting from a control simulation (*ctrl*) performed at constant $CO_2$ concentration, three transient simulations forced by rising $CO_2$ concentrations are performed. In the first of those transient simulations (*conc*) only the carbon-concentration effect is active, which means that the rising $CO_2$ concentration is "seen" only by the photosynthesis code of the model, while the radiation code constantly "sees" the $CO_2$ value of the control simulation. Conversely, in the second transient simulation (*clim*) only the carbon-climate effect is active, i.e. only the radiation code "sees" the rising $CO_2$ concentrations but not the photosynthesis model. In the third simulation (*full*) both effects are simultaneously active. These simulations are run once for LGM and once for PI conditions. – In the following, we will use the term 'experiment' to refer to one of the two cases LGM or PI. 'Simulation' will refer to one of the four model runs *ctrl*, *clim*, *conc* or *full*.

The CO$_2$ concentrations for the *ctrl* simulations of the two experiments are 185 ppm (LGM) and 285 ppm (PI), which are also the initial conditions for the respective transient simulations. Experiments were performed with MPI-ESM (see below). In fact, we performed only the LGM experiment for this study since we could use published MPI-ESM CMIP5 simulations (called *piControl*, *esmFdbk1*, *esmFixClim1* and *1pctCo2*) for our purpose that were performed for PI conditions with the same

model version. The LGM simulations were initialized from restart files of the MPI-ESM CMIP5 last glacial maximum spin-up experiment (1800 simulation years long), extended by another 200 years with dynamic vegetation now switched on. The PI simulations used for our study were initialized from a spin-up experiment covering more than 3000 years. For the transient simulations *clim*, *conc* and *full*, the same atmospheric CO$_2$ concentration increase is imposed over a period of 150 years in both experiments (see Fig. 1), acting differently in the three simulations as explained above. The forcing for our LGM

experiment is obtained by reducing the standard PI CO$_2$ forcing by 100 ppm to account for lower glacial CO$_2$ concentrations while preserving the rate of change. Because CO$_2$ concentrations thereby increase by the same amount, the different reaction of the Earth system to the CO$_2$ rise in the two experiments should mostly be attributable to the different initial conditions, i.e. the glacial-interglacial atmospheric CO$_2$ offset and the particular characteristics of the initial climates. The distribution of ice sheets is prescribed to the appropriate LGM and PI conditions and is kept constant in all simulations.

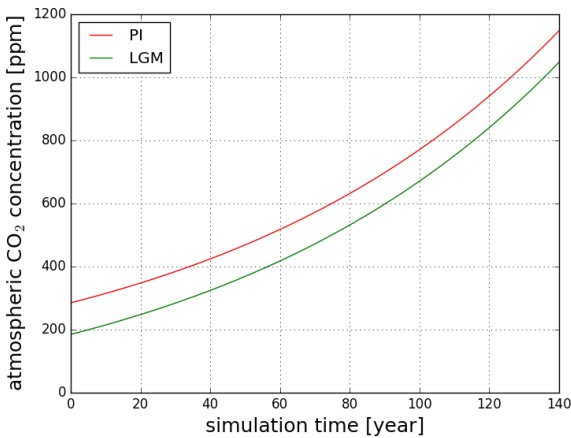

**Figure 1.** CO$_2$ change scenarios as prescribed for the LGM and PI experiments: Starting from 185ppm ("Last Glacial Maximum", green line) and starting from 285ppm ("Pre-Industrial", red line).

The experiments are conducted with the Earth-System Model of the Max Planck Institute (MPI-ESM) using the version described in Giorgetta (2013). The MPI-ESM consists of the atmosphere component ECHAM6 and the ocean component MPIOM, both including submodels for simulating the land and ocean carbon cycles. Because atmospheric CO$_2$ concentrations are prescribed in our experiments, the oceanic and terrestrial carbon cycles are decoupled so that changes in the ocean carbon cycle are irrelevant for terrestrial carbon reservoirs that are of main interest here; nevertheless oceanic carbon fluxes play a

role for calculating the overall carbon cycle feedback in our study and the physical ocean remains an important component of the climate dynamics affecting also the land carbon cycle. The land component JSBACH comprises the DYNVEG model

for simulation of natural changes in the geographical distribution of vegetation controlled by competition and wind and fire disturbances (Reick et al., 2013), and the BETHY model (Knorr, 2000) for simulation of the fast biochemical and biophysical processes of the biosphere, in particular photosynthetic production that is simulated following the Farquhar model (Farquhar et al., 1980) for C3 and the Collatz model (Collatz et al., 1992) for C4 plants. Vegetation is represented by eight plant functional types that differ in phenology and physiology and interact dynamically (see Brovkin et al. (2013) for an evaluation of the present implementation of dynamic biogeography). There is no anthropogenic land cover change considered in the experiments here. Terrestrial carbon dynamics are calculated with CBALANCE (Reick et al., 2010), representing vegetation, litter, and soils by seven carbon pools, where temperature dependence of heterotrophic respiration is modeled by a Q10-formula and turnover rates are in addition dependent on soil humidity. The oceanic biogeochemistry model HAMOCC (Ilyina et al., 2013) calculates sea-air gas exchange, water column processes and sediment dynamics. $CO_2$ exchange between sea and air is calculated with a temperature dependent rate based on the thermodynamic disequilibrium at the interface. Carbon is then cycled as organically fixed carbon, dissolved inorganic carbon and calcium carbonate in the water column and sediments. Temperature, nutrient and light dependent biological cycling of carbon within the water column is represented by an extended NPZD model (Six and Maier-Reimer, 1996), inorganic carbon cycling is based on Maier-Reimer and Hasselmann (1987), using updated chemical constants by Goyet and Poisson (1989).

## 3 Analysis framework

Here we introduce the mathematical framework for analyzing our simulations in the next sections. First we describe how we apply the factor separation method by Stein and Alpert (1993) to separate the relative contributions of the carbon-concentration and carbon-climate effects to the overall changes in terrestrial carbon reservoirs. In the remainder of the section we describe the mathematical framework to disentangle the oceanic and atmospheric contributions to the overall carbon cycle feedback, as well as the contributions of those two effects to the feedback. This feedback framework was originally introduced by Friedlingstein et al. (2003) and further discussed by Gregory et al. (2009). We apply it here in the variant with prescribed atmospheric $CO_2$ (Ciais et al., 2013, Box 6.4).

We apply the factor separation method of Stein and Alpert (1993) as follows. Let $C_L$ denote the total land carbon. The pure effects of the carbon-concentration and carbon-climate effects are individually quantified by the differences

$$
\begin{aligned}
\Delta C_{L,conc}(t) &:= C_{L,conc}(t) - \overline{C}_{L,ctrl} \\
\Delta C_{L,clim}(t) &:= C_{L,clim}(t) - \overline{C}_{L,ctrl}
\end{aligned}
\tag{1}
$$

where the indices at the right hand side $C_L$-values refer to the simulations from which the values were obtained, while the indices to the $\Delta C_L$-values at the left hand side refer to the effect considered. The time dependence $t$ appears only for the values from transient simulations, but not for values from the control simulations which enter our calculations as mean values (indicated as a bar over the symbol). In addition, we quantify the 'synergy' between the carbon-concentration and the carbon-climate effects, which is that part of the land carbon storage difference between the *full* and *ctrl* simulation that cannot be

explained by a linear addition of the individual effects:

$$\Delta C_{L,syn}(t) \quad := \quad (C_{L,full}(t) - \overline{C}_{L,ctrl}) - (\Delta C_{L,conc}(t) + \Delta C_{L,clim}(t)). \tag{2}$$

Note that in this way all separate factors sum up to the land carbon change in the *full* simulation:

$$\Delta C_{L,full}(t) \quad = \quad \Delta C_{L,conc}(t) + \Delta C_{L,clim}(t) + \Delta C_{L,syn}(t). \tag{3}$$

For the feedback analysis we consider the following differences in near surface temperature and atmospheric $CO_2$ concentration that develop in the transient simulations:

$$\begin{aligned} \Delta T_{clim}(t) \quad &:= \quad T_{clim}(t) - \overline{T}_{ctrl} \\ \Delta cc(t) \quad &:= \quad cc(t) - cc_{ctrl}. \end{aligned} \tag{4}$$

The concentration of atmospheric $CO_2$ is denoted here by "$cc$" and measured in ppm $CO_2$. Since $cc(t)$ is the same for all transient simulations of a particular experiment, the index specifying the simulation has been omitted. With these definitions one can now introduce the two land carbon sensitivities

$$\begin{aligned} \beta_L(t) \quad &:= \quad \frac{\Delta C_{L,conc}(t)}{\Delta cc(t)} \\ \gamma_L(t) \quad &:= \quad \frac{\Delta C_{L,clim}(t)}{\Delta T_{clim}(t)}. \end{aligned} \tag{5}$$

$\beta_L$ [PgC/ppm] measures how strongly land carbon is affected in the *conc* simulation by changes in atmospheric $CO_2$; since in the *conc* simulation only the carbon-concentration effect is active, $\beta_L$ measures the strength of this effect alone. Analogously, $\gamma_L$ [PgC/K] measures how strongly land carbon is affected by temperature changes in the *clim* simulation; because in this simulation only the carbon-climate effect is active, it represents the strength of this effect alone. Similar sensitivities can be defined for ocean carbon but they will not be needed in this study.

In addition to $\beta_L$ and $\gamma_L$ we will need below the sensitivity of temperature to increasing $CO_2$ concentrations in our simulations, known as temperature sensitivity [K/ppm] (Friedlingstein et al., 2003):

$$\alpha(t) \quad := \quad \frac{\Delta T_{clim}(t)}{\Delta cc(t)}. \tag{6}$$

Note that in this framework $\alpha$, $\beta_L$ and $\gamma_L$ are time dependent – a point that will be further discussed below.

To introduce a measure for the feedback strength, the global carbon balance needs to be considered. Since the $CO_2$-concentration is prescribed in our simulations, atmospheric carbon is not affected by ocean-atmosphere or land-atmosphere carbon fluxes, i.e. the global carbon budget is not closed. But one can diagnose how much external $CO_2$ emissions into the atmosphere would be needed to close the global carbon budget. Considering our full simulation, the prescribed change in atmospheric carbon must match the imagined external carbon emissions $I_{ext}(t)$ minus the carbon uptake by ocean and land $\Delta C_{OL,full}(t)$:

$$\Delta C_A(t) \quad = \quad I_{ext}(t) - \Delta C_{OL,full}(t). \tag{7}$$

Assuming that ocean and land carbon uptake are proportional to the increase in atmospheric $CO_2$, one can define the proportionality factor $f(t)$ by

$$\Delta C_{OL,full}(t) \quad =: \quad -f(t)\Delta C_A(t) \tag{8}$$

where the reason for introducing here a minus sign will get clear below. With this one obtains from (7)

$$\Delta C_A(t) = A(t)I_{ext}(t), \quad \text{with} \quad A(t) = \frac{1}{1 - f(t)}. \tag{9}$$

$A(t)$ is called the airborne fraction (compare e.g Gregory et al. (2009)). If atmospheric carbon content would not be prescribed, $A(t)$ would describe how much of the carbon $I_{ext}(t)$ added to the atmosphere would remain in it. Following Roe (2009), from
the viewpoint of feedback analysis $A(t)$ is the 'gain' of the feedback: For $A(t)$ larger/smaller 1 the feedback is positive/negative, i.e. the forcing $I_{ext}(t)$ induces via (8) additional carbon fluxes into/out of the atmosphere. By (9) the gain of the feedback is completely determined by the value of $f(t)$, which – also following Roe (2009) – is called the 'feedback factor'. Note that the sign in eq. (8) is chosen such that a positive/negative feedback corresponds to a positive/negative sign of $f(t)$.

In the present study we focus on the terrestrial contribution to the carbon cycle feedback. This contribution is obtained as
follows. Splitting $\Delta C_{OL,full}(t)$ in (8) into the separate contributions $\Delta C_{L,full}(t)$ from land and $\Delta C_{O,full}(t)$ from ocean, one can define individual land and ocean feedback factors

$$\begin{aligned}
\Delta C_{O,full}(t) &=: -f_O(t)\Delta C_A(t) \\
\Delta C_{L,full}(t) &=: -f_L(t)\Delta C_A(t)
\end{aligned} \tag{10}$$

so that

$$f(t) \quad = f_O(t) + f_L(t). \tag{11}$$

Hence the individual feedback factors from ocean and land contribute additively to the global feedback factor.

To disentangle the contributions of the carbon-concentration and the carbon-climate effect to $f_L(t)$, we assume that the synergy term in (3) is small compared to the others. Then one can express the carbon change in the *full* simulation induced by the combined action of the two effects by summing the carbon changes induced by the individual effects diagnosed in the simulations *conc* and *clim*. Using the definitions for $\alpha$, $\beta_L$ and $\gamma_L$ from above, and noting that atmospheric carbon content and
atmospheric $CO_2$ concentration are related via the conversion factor $m = 2.12 \text{ Pg/ppm}$ (Flato et al. (2013), page 471), one thus finds

$$f_L(t) \quad = \quad -\frac{\alpha(t)\gamma_L(t) + \beta_L(t)}{m}. \tag{12}$$

Here the first term quantifies the contribution from the carbon-climate effect, while the second that from the carbon-concentration effect.

Please note that the feedback considered here is different from that originally considered by Friedlingstein et al. (2003) or in the C[4]MIP study (Friedlingstein et al., 2006): Besides the fact that we focus on the feedbacks induced by terrestrial

processes only, the more important difference to our study is that Friedlingstein et al. (2003) considered only the feedback induced by the carbon-climate effect (see Friedlingstein et al. (2003, eq. (8b)), Friedlingstein et al. (2006, eq. (1)), or Gregory et al. (2009, eq. (17))), while in our study, following Gregory et al. (2009, eq. (16)), we quantify the feedback induced by the carbon-climate *and* carbon-concentration feedback together (see our eq. (9)). Please note also that there is a confusion in the literature concerning the names 'gain' and 'feedback factor'; in our study we follow the naming convention of Roe (2009), who made aware of this confusion.

## 4    Comparison of the simulated LGM and PI equilibrium states

Here we compare key climate and carbon variables from the LGM and PI *ctrl* simulations that are the initial states for the transient simulations analyzed in the next sections. Globally, mean near surface temperatures are 4.5 K colder in the LGM state than in the PI state but locally, temperatures differ by 20 K and more (see Fig. 2). Compared to PI, more water is available for vegetation growth in the LGM state, especially in the tropics and subtropics. This plant water availability is measured here in terms of the relative amount of water above wilting point in the root zone of the soil, a value of 1 indicating optimal moisture levels and 0 indicating that photosynthesis is inhibited by water scarcity. Inland glaciers extend throughout most of North America and northern Europe in the LGM state and the sea level is considerably lower, leading to a different geography, especially in the Bering Strait and the Malay Archipelago. On global scale, less area is covered by vegetation and dense vegetation is restricted to the tropical zone (compare Fig. 3). In the PI state, vegetation reaches far more into the extratropics and the mid latitudes are more densely covered by vegetation. Terrestrial carbon reservoirs are larger in the PI experiment almost everywhere (see Fig. 3). Globally, terrestrial carbon reservoirs contain 1986 PgC in the LGM and 3041 PgC in the PI state. Our difference in carbon storage (1055 PgC) matches the difference of 1030±625 PgC in non-permafrost land carbon obtained by Ciais et al. (2012) from combining model simulations with carbon and oxygen isotope data from sediment and ice cores; note that changes in permafrost carbon are not part of our simulations.

## 5    Reaction of the Earth system to rising $CO_2$ concentration under different boundary conditions

The climate system reacts differently to rising $CO_2$ concentrations under LGM and PI boundary conditions. Fig. 4 shows changes in global mean near surface temperature and plant water availability in the transient simulations. Due to rising $CO_2$ concentrations, global mean near surface temperature increases in the *clim* and *full* simulations while plant water availability decreases. Both of these changes are larger in the LGM experiment. The similarity of temperature changes in the *clim* and *full* simulations shows that the carbon-concentration and synergistic effects do not considerably affect global mean near surface temperature. Nevertheless, also the carbon-concentration effect creates a small global warming towards the end of both experiments, as can be seen from the curves of the *conc* simulations. Gregory et al. (2009) explained this by less evapotranspiration under increased $CO_2$ concentrations. The radiative effect of increased stomatal closure has been shown by previous studies, e.g. Doutriaux-Boucher et al. (2009). The influence of the carbon-concentration effect on other physical variables, however,

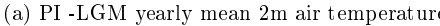

(a) PI -LGM yearly mean 2m air temperature      (b) PI - LGM plant water availability

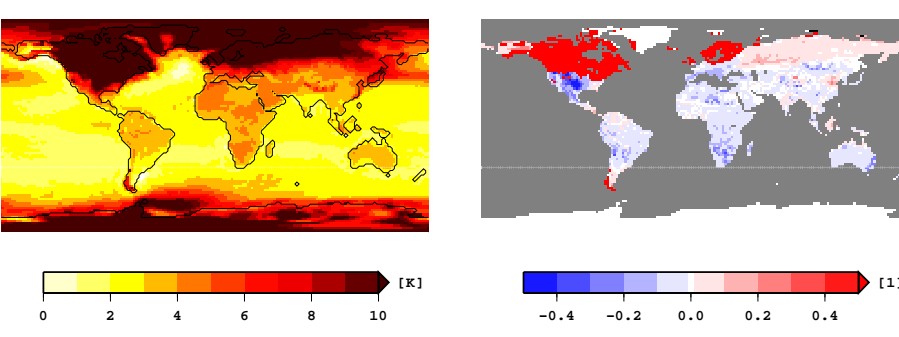

**Figure 2.** Differences between the LGM and PI climates obtained in the respective *ctrl* simulations: a) Difference in global mean near surface temperatures and b) difference in plant water availability. Here the values in the LGM state are substracted from the values in the PI state. Land areas that are covered by ice in the LGM but not in the PI equilibrium state show soil humidity differences > 0.4.

is more important for the terrestrial carbon dynamics. For example, plant water availability, the second most important environmental constraint on most terrestrial carbon fluxes in the model, rises in the global average due to increased water use efficiency in connection with the carbon-concentration effect and decreases due to higher evapotranspiration losses under the higher temperatures as a consequence of the carbon-climate effect. Climate change dominates plant water availability changes in the *full* simulation, but also a clear influence of the carbon-concentration effect and their synergies on plant water availability is apparent.

Figure 5 shows the change of terrestrial carbon storage in the transient simulations. Overall, the carbon-concentration effect increases terrestrial carbon storage in response to the rising $CO_2$ concentration in both experiments (see the curves $\Delta C_{L,conc}$). This effect is stronger in the LGM than in the PI experiment. Carbon reservoir changes due to the carbon-climate effect are negative and of similar size in the two experiments (see curves $\Delta C_{L,clim}$). In both experiments, synergies of the two effects are small in the global integral (see curves $\Delta C_{L,syn}$). This shows that linear additivity of the carbon-climate and carbon-concentration effects can be assumed on the global scale for our experiments, even for the large climate perturbations considered here. This is important in the following because by this additivity one can separate the individual contributions of the two effects to the feedback strength by means of eq. (12) (see the discussion there).

From Fig. 5 it becomes clear that the same absolute increase in atmospheric $CO_2$ concentration triggers different reactions of terrestrial carbon storage in corresponding simulations of the LGM and PI experiments. This is also reflected in the terrestrial carbon cycle sensitivities as shown in Fig. 6 where the sensitivity values for the LGM and PI experiments are presented as a function of simulation time. In the following, before discussing the strength of the carbon cycle feedback, first the sensitivities and their temporal development will be studied separately.

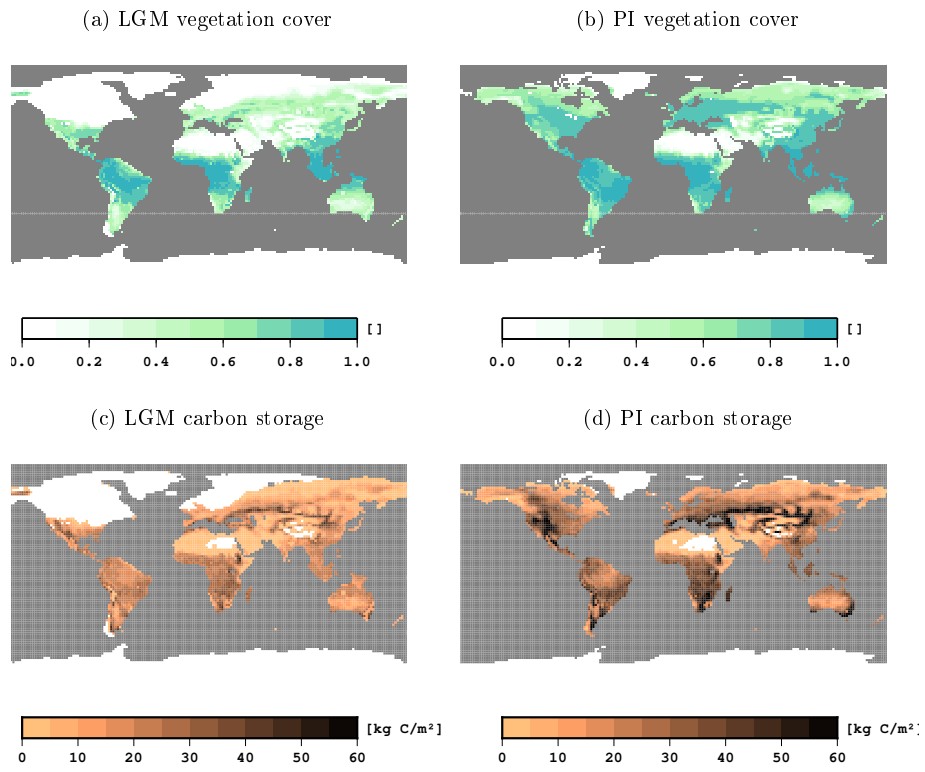

**Figure 3.** Vegetation cover and carbon storage in the LGM and PI *ctrl* simulations. Vegetation cover is given as fraction of grid cell covered with vegetation, and carbon storage in $kgC/m^2$.

## 5.1 The carbon-concentration effect

Initially $\beta_L$ increases in both experiments but the increase is steeper under glacial conditions. This stronger carbon-concentration effect in the LGM experiment is mostly due to the lower $CO_2$ concentrations: In both experiments, photosynthesis is initially carboxylation rate limited. In other words, in both experiments the fraction of available radiative energy that the plants are able to use to build up organic matter is initially limited by low atmospheric $CO_2$ concentrations. This initial $CO_2$ limitation is lifted by increasing $CO_2$ concentrations, which leads to increasing primary productivity that allows for extension of vegetation and increasing terrestrial carbon storage. This mechanism becomes obvious from Fig. 7, which shows the dependence of primary production rate on $CO_2$ concentration calculated directly from the equations for C3 photosynthesis, which dominates global natural productivity, implemented in JSBACH. At low ambient $CO_2$ concentrations, productivity increases steeply with rising $CO_2$ but its sensitivity gets smaller at higher $CO_2$ concentrations due to the convex nature of the underlying functionality. In our experiments the carbon-concentration effect on productivity differs most substantially in the tropics, where temperatures are similar but the lower LGM ambient $CO_2$ concentration makes productivity more sensitive to $CO_2$ increases in the glacial

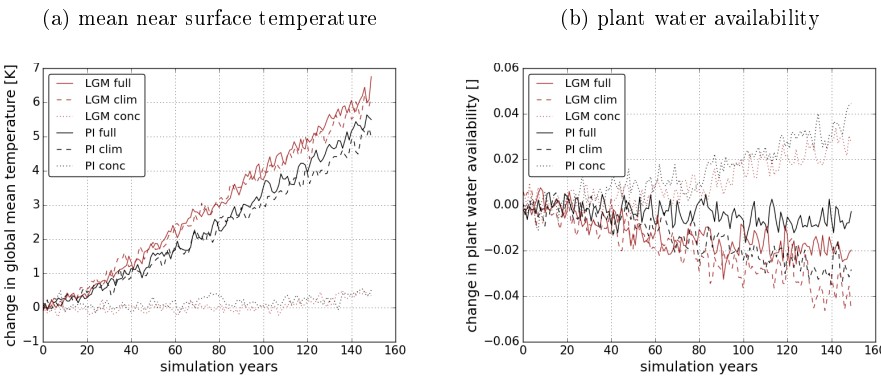

**Figure 4.** Climatic changes in the *full* simulation (continuous lines), *clim* simulation (dashed lines) and the *conc* simulation (dotted lines) due to rising $CO_2$ concentrations in the LGM experiment (red) and PI experiment (black). a) shows the globally averaged change in near surface temperature and b) in plant water availability.

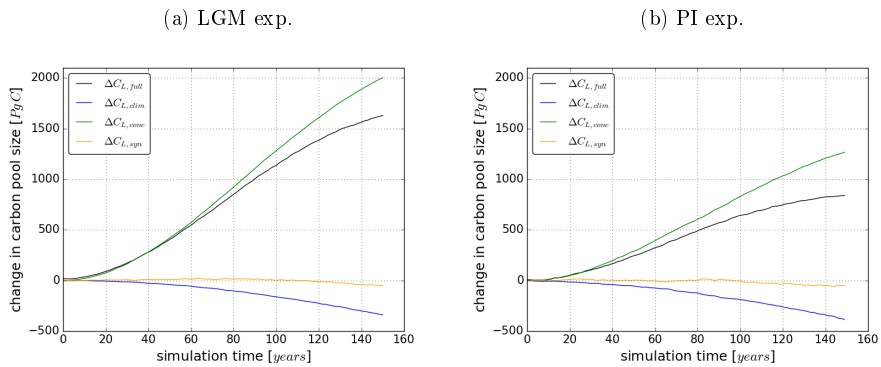

**Figure 5.** Change in terrestrial carbon storage [PgC] in the *full* simulations (black curves) and split into factors (coloured curves) as computed from eqs. (1) and (2) for (a) the LGM experiment and (b) the PI experiment.

setting. Additionally, vegetation has more room to expand and can generally grow denser in the glacial tropics than under the drier pre-industrial conditions where tropical forests are more regularly perturbed by wild fires.

Fig 6a shows that, after 30 to 40 years, the increase of $\beta_L$ slows down and its values eventually start to decrease. Arora et al. (2013) attribute this behaviour to the different response time of primary production and biomass decomposition. While productivity increases almost instantaneously with rising $CO_2$ concentration, biomass decomposition initially remains unchanged and increases only when after a temporal delay of the order of the lifetime of plants the additional carbon from higher plant productivity reaches the litter and soil carbon reservoirs. Additionally the carbon-concentration effect becomes less effective at high productivity levels because carbon density of living vegetation is reaching upper limits. In fact, the amount of carbon allocatable to biomass carbon reservoirs is limited in JSBACH to account for a down regulation of productivity in mature vegetation.

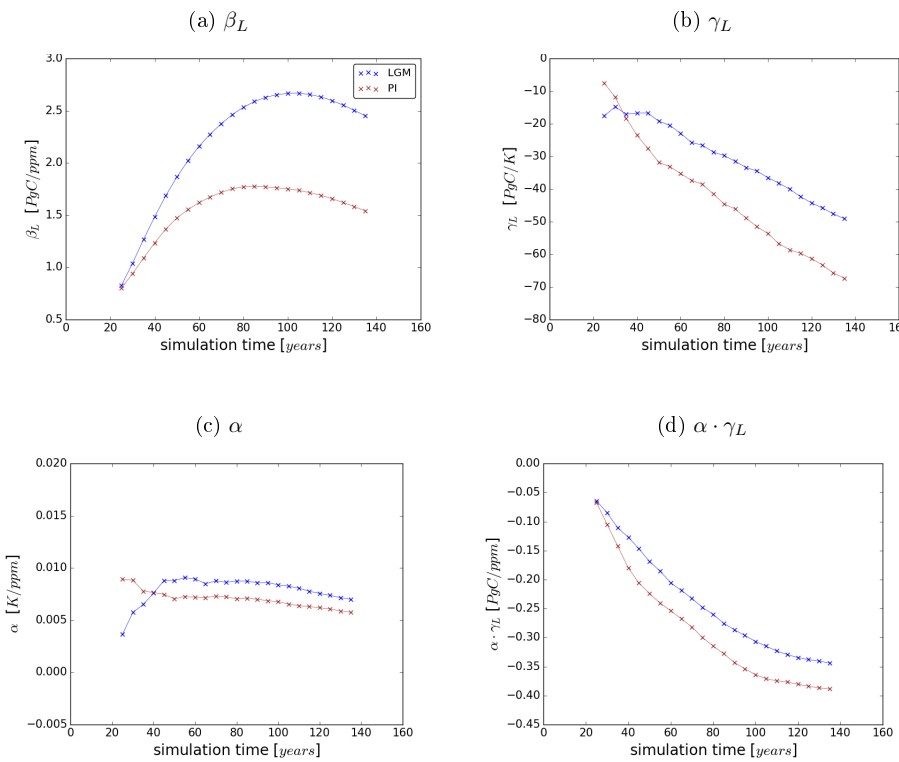

**Figure 6.** Sensitivities $\beta_L$ and $\gamma_L$ to the carbon-concentration and carbon-climate effect (respectively) and temperature sensitivity $\alpha$ in the LGM (blue) and the PI experiment (red). Values are computed as a 20 year average around the indicated data point.

But also the sensitvity of productivity to ambient $CO_2$ changes : Fig 7 shows a transition point from high to low dependence on $CO_2$ changes. Below the transition point photosynthesis is carboxylation rate limited, while beyond the transition point it is limited by lack of radiation (see any textbook on photosynthesis). Accordingly, as long as $CO_2$ availability stays to be the main limitation for productivity, the carbon-concentration effect of rising $CO_2$ concentration leads to large increases in productivity. In our experiments, the prescribed $CO_2$ concentration rise is however large enough to reach a point where insolation becomes more limiting to productivity than $CO_2$ availability. From that transition point on, the effectivity of the carbon-concentration effect is saturating. In the PI experiment ambient $CO_2$ concentration reaches that point of saturation earlier than in the LGM experiment, leading to a shorter period in the PI experiment where primary productivity is limited by $CO_2$ availability and thus highly sensitive to rising $CO_2$ concentrations.

## 5.2 The carbon-climate effect

The sensitivity $\gamma_L$ grows increasingly negative in both experiments (see Fig. 6b) and increasingly larger in absolute value in the PI experiment than in the LGM experiment. Although $\gamma_L$ values are clearly different in the two experiments, the overall

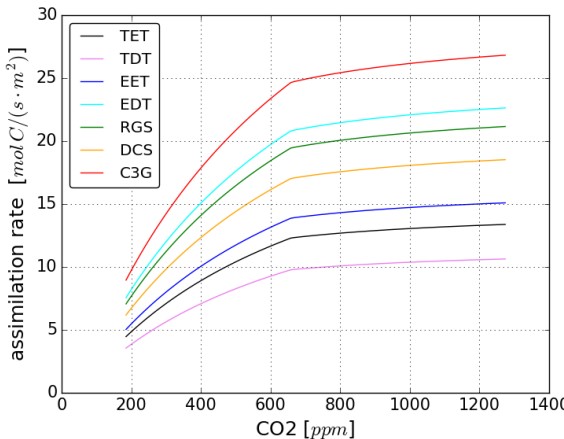

**Figure 7.** Dependence of gross assimilation per m$^2$ leaf area on ambient CO$_2$ concentration at 20°C leaf temperature according to the implemented photosynthesis model (Farquhar et al., 1980) for C3 plant physiology. Abbreviations stand for individual vegetation types: TET for tropical evergreen trees, TDT for tropical deciduous trees, EET for extratropical evergreen trees, EDT for extratropical deciduous trees, RGS for raingreen shrubs, DCS for deciduous shrubs, C3G for C3 grasses.

terrestrial carbon reservoir changes in the *clim* simulations, from which the $\gamma_L$ values are computed (see eq. (5)), are almost similar (compare Fig. 5). The reason for this is that also the temperature sensitivity $\alpha$ varies between the two experiments. Throughout the simulation $\alpha$ is larger in the LGM experiment. The higher temperature sensitivity and the lower carbon cycle sensitivity $\gamma_L$ partially compensate differences between the PI and LGM cases as is seen from Fig. 6 (d) where the product $\alpha\gamma_L$ is plotted; it is this combination of sensitivities that determines the strength of the carbon-climate effect (compare equation

(12)). Thereby the carbon-climate effect differs much less between the LGM and PI case than the carbon-concentration effect discussed above.

     To understand the processes behind the different $\gamma_L$ sensitivity in the two experiments, it is useful to analyze first how climate change induces carbon losses differently in the tropics and extratropics. Table 1 lists the change in soil respiration $\Delta$Rh and net primary productivity $\Delta$NPP per degree temperature change as well as their ratio separately for tropics and extratropics in the

two *clim* simulations. In both simulations this ratio is smaller than 1 in the tropics (carbon fluxes into land reservoirs change more than fluxes into the atmosphere) but larger than 1 in the extratropics (carbon fluxes into the atmosphere change more than fluxes into land carbon reservoirs), indicating a very different reaction of the carbon cycle under climate change in these two regions. In the tropics, net primary productivity and soil respiration decrease (see table), indicating that living conditions deteriorate. This has two reasons: Firstly, it gets drier so that plant productivity and also soil decomposition decrease. Secondly,

the already hot tropical climate is getting even hotter so that physiological limitations are reached more frequently, deteriorating plant productivity by damaging the photosynthetic apparatus (implemented as 'heat inhibition' in JSBACH). The reduction in NPP is much larger than the reduction in soil respiration, hence in the tropics land carbon losses are mostly driven by reduced plant productivity. In the extratropics the situation is different: values of NPP and soil respiration (see table) both rise under

the warming climate because physiological processes speed up. But since ultimately soil respiration is fed from NPP, the considerably larger increase in soil respiration cannot be a result of the enhanced carbon input. The explanation, instead, is enhanced decomposition of soil carbon that had accumulated in those vast cold boreal areas already in the control simulation from which the transient simulations are initialized. Hence in the extratropics land carbon losses are mostly driven by enhanced soil respiration of 'old' carbon.

Having identified the major drivers for carbon losses in the tropics and extratropics, one can now understand why the sensitivity $\gamma_L$ is larger in the PI than in the LGM experiment. In the tropics reduced plant productivity is the major driver, and productivity is sensitive in the PI than the LGM experiment (see table 1) because growth conditions deteriorate from already initially drier and hotter levels. In the extratropics enhancement of soil respiration was found to be the major driver, and soil respiration reacts more sensitive in the PI than in the LGM experiment (see table 1) because vegetation extends much farther

north under the warmer conditions and in absence of ice sheets, going along with vastly more extratropical 'old' soil carbon. Hence both in the tropics and in the extratropics the land carbon cycle is more sensitive to climate change in the PI experiment.

    While our model setup allows to study the reaction of active carbon reservoirs to perturbations, it does not include inert carbon reservoirs which could be activated under a strong forcing (i.e. permafrost soils). This might be particularly important for the comparison of $\gamma_L$ between the LGM and the PI state since Ciais et al. (2012) estimate that there was a considerably larger

amount of inert carbon stored on land at the LGM than in the Holocene. Therefore, it has to be stressed that the sensitivities found in this study do only consider active carbon reservoirs.

**Table 1.** Sensitivity of net primary productivity NPP and soil respiration Rh to the carbon-climate effect. These sensitivities ($\Delta$NPP/$\Delta$T and $\Delta$Rh/$\Delta$T) are computed from the *clim* simulation by first integrating NPP and RH over the particular region (tropics, extratropics) and over the full simulation period and then dividing by the temperatur change in this region. $\Delta$Rh/$\Delta$NPP is the quotient of the two sensitivities. 'Tropics' refers here to the latitudinal belt between 30° South and 30° North and 'extratropics' to the remaining part of the globe. Here, $\Delta$NPP and $\Delta$Rh are considered positive for plant carbon uptake and soil carbon loss, respectively.

| sensitivity [Pg C/K] | tropics | | extratropics | |
|---|---|---|---|---|
| | LGM | PI | LGM | PI |
| $\Delta$NPP/$\Delta$T | -134.6 | -151.2 | 10.8 | 28.6 |
| $\Delta$Rh/$\Delta$T | -55.9 | -49.7 | 17.1 | 48.2 |
| $\Delta$Rh/$\Delta$NPP | 0.42 | 0.33 | 1.59 | 1.69 |

## 5.3   Feedback strength of the terrestrial carbon cycle

The carbon-climate and the carbon-concentration effect cause a feedback of the terrestrial carbon cycle to rising atmospheric $CO_2$ concentrations. The constantly negative values of the strength $f_L$ of this feedback (see Fg. 8) demonstrate that it dampens

the effect of the forcing so that less carbon is left in the atmosphere than emitted. Accordingly, the feedback is negative in both experiments. From the beginning of the simulations, the feedback strength grows increasingly negative in both experiments, a trend that reverses later on with an earlier minimum in the PI experiment. This reflects the different development of $\beta_L$ that

dominates the feedback strength for both PI and LGM (compare values of $\beta_L$ and $\alpha\gamma_L$ in Fig. 6). The dominance of $\beta_L$ is particularly visible towards the end of the simulations, where the timing of the reversal of the trends in $f_L$ match those in $\beta_L$ (compare fig. 6). The constantly higher absolute values of $f_L$ in the LGM setting show that the feedback is much stronger

under LGM conditions, especially towards the end of the simulations. Because $\beta_L$ is dominating $f_L$, the stronger terrestrial LGM feedback is also explained by the mechanisms identified in section 5.1 to cause the higher LGM sensitivity to the carbon-concentration effect.

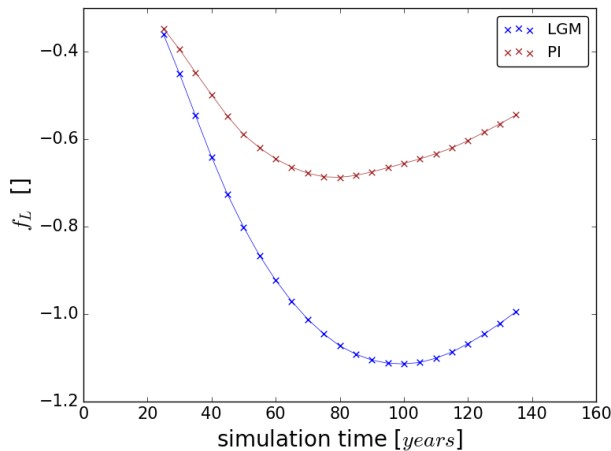

**Figure 8.** Feedback strength $f_L$ computed from eq. (12) for the terrestrial carbon carbon cycle in the LGM and the PI experiment.

## 6   Discussion and Conclusion

In the present study we investigated in simulations how the terrestrial carbon cycle feedback differs between pre-industrial

(PI) times and during the last glacial maximum (LGM). This was done by separating the contributions from the carbon-concentration and carbon-climate effects that induce this feedback in $C^4$MIP type simulations. These simulations starting either at PI or LGM conditions are rather artificial, since the $CO_2$ forcing scenario used to probe the feedbacks neither resembles the atmospheric $CO_2$ changes during the holocene, nor is it realistic for recent times (compare Fig. 1). But they are not meant to be historically realistic. Instead, such artificial scenarios have been introduced to facilitate the comparison of the carbon

cycle feedback across different models (Gregory et al., 2009). In our study we adopted this approach for a comparison of this feedback between different climate states.

An important question for the applicability of the $C^4$MIP type feedback analysis is the additivity of the two effects for global land carbon storage because only then the feedback strength can be properly split into separate contributions from the two effects (see (Gregory et al., 2009) and our discussion in section 3). Our factor separation analysis (Stein and Alpert, 1993)

revealed that their synergy is rather small for both the PI and LGM case, meaning that we can indeed consider the two effects

independently to understand the simulated feedback behaviour. Concerning this additivity models seem to behave differently: Gregory et al. (2009) reported significant deviations from additivity for the HadCM3LC model.

Generally, the values of the carbon sensitivities $\beta_L$ and $\gamma_L$ are time dependent (compare Fig. 6), but for easier comparison they are usually reported taking their values at the end of the simulation period (see e.g. (Ciais et al., 2013)). The respective values from our simulations are given in table 2, together with their CMIP5 intermodel range. Since we used for the analysis of PI conditions the same data from MPI-ESM that entered the CMIP5 study by Arora et al. (2013), one should expect that published values for $\beta_L$ and $\gamma_L$ should be similar. This is indeed true for $\beta_L$ for which we find 1.42 PgC/ppm while (Arora et al., 2013, table 2) find 1.46 PgC/ppm. But for $\gamma_L$ we calculate -68.6 PgC/K while they report -83.2 PgC/K. We attribute this apparent inconsistency to differences in the way we and Arora et al. (2013) compute sensitivities: While we use as reference mean values from the control simulation (see eqs. (1) and (4)), we guess that Arora et al. (2013) use as reference the value from the first year of the respective transient simulation. Thereby the resulting sensitivity values are not only sensitive to random climate variations at the end of the simulations (which are typically smaller than changes from the strong forcing), but also sensitive to such variations at their begin. For the considered sensitivities, this effect should be largest for temperature that is varying at much shorter time scales than carbon stocks. Accordingly, $\beta_L$ values should be less sensitive to the way they are computed and this may explain why our $\beta_L$ values are similar, but $\gamma_L$ values differ.

For our further considerations it is interesting to see how our LGM carbon sensitivities relate to published PI values. In view of the technical complications just mentioned, such a comparison makes sense only for $\beta_L$. We see from table 2 that our LGM $\beta_L$ is considerably larger than the PI value of any CMIP5 model. This may be taken as an indication that our result for differences in $\beta_L$ between PI and LGM is even robust against uncertainties in representing climate and carbon cycle in models. Since, as we discussed in section 5.3, the terrestrial feedback strength, as measured by the feedback factor $f_L$, is dominated by the contribution from $\beta_L$ (compare also eq. (12)), it is clear that for LGM and PI the feedback is dominated by the carbon-concentration effect. Hence, also the much larger LGM feedback factor $f_L$ – almost twice the PI value – should be a robust result from our study.

**Table 2.** Terrestrial carbon sensitivities $\beta_L$ and $\gamma_L$, associated feedback factor $f_L$, as well as the global feedback factor $f$ that includes the oceanic feedback (see eq. (11)) from our simulations for PI and LGM, as well as their published CMIP5 model range for PI. Our values (columns PI and LGM) are taken as their value after 140 years of simulation. The CMIP5 model range is taken from Arora et al. (2013), considering only models without nitrogen cycle. The CMIP5 ranges for $f_L$ and $f$ have been computed using the published CMIP5 sensitivities in eq. 12 and its ocean analogue together with eq. 11. Because the intermodel range for $\alpha$ is not given in Arora et al. (2013), $\alpha$s were calculated from the gain $\hat{g}_E$ provided in Arora et al. (2013)'s Fig. 9.

|  | LGM exp. | PI exp. | CMIP5 |
|---|---|---|---|
| $\beta_L$ [$PgC/ppm$] | 2.19 | 1.42 | 0.74 – 1.46 |
| $\gamma_L$ [$PgC/K$] | -53.0 | -68.6 | -30.1 – -88.6 |
| $f_L$ | -0.87 | -0.48 | -0.07 – -0.48 |
| $f$ | -1.27 | -0.81 | -0.42 – -0.85 |

So far, we have concentrated our study on the terrestrial part of the Earth system, but it is interesting to consider for a moment also the oceanic contributions to the feedback to discuss the relevance of our results for the carbon cycle feedback in the Earth system as a whole. Our simulations have been performed also with the ocean carbon cycle being active. Accordingly, one can calculate from our simulations also the ocean feedback factor $f_O$ (see eq. (10)). A basic property of the global feedback strength is that ocean and land contributions to the overall feedback factor $f$ are additive (compare eq. (11)). Obtaining in this way the global feedback strength, one sees from the values in table 2 that in our simulations the terrestrial component dominates the global feedback in the LGM case, while both contributions are of approximate equal size for pre-industrial climate.

As discussed in section 5.1 and 5.2, the difference in carbon sensitivities between the LGM and the PI experiments comes mostly from the different initial conditions of these experiments. But there is also a strong dependence on the strength of the $CO_2$ forcing. For example, the difference in $\beta_L$ depends largely on whether the $CO_2$ reaches values high enough to produce a switch from carboxylation limited assimilation to radiation limited assimilation. Additionally, bioclimatic limits of vegetation, model specific maximum productivity rates, the choice of the global value for the wilting point and the assumed maximum vegetation density introduce limitations to the system that shape the behaviour of terrestrial carbon storage in the model. Such limitations should also exist in reality but are hard to quantify.

Besides the dependence on the forcing scenario, the calculated sensitivity parameters are also time dependent. This is due to the fact that the Earth system's response to the imposed forcing is not entirely instantaneous. Many physical and biogeochemical processes react on longer timescales (e.g. plant and ecosystem growth and inertia in heat and carbon reservoirs), which also interact and thereby complicate the system's response. This simultaneous dependence of the $\alpha$, $\beta$ and $\gamma$ values on system state and forcing is well known (Gregory et al., 2009; Arora et al., 2013). Accordingly, these sensitivity metrics do not characterize an Earth system state as such, but only a combination of initial state and forcing scenario. Hence to isolate their state dependence one must consider simulations with similar forcing. This is is the reason why in our study we subjected the LGM and PI state to the same increase in $CO_2$.

To conclude, the present study has demonstrated that $C^4MIP$ type simulations can be used to understand why the Earth system may react differently to rising $CO_2$ concentrations under LGM and PI conditions. In the two experiments performed here for LGM and PI conditions, the terrestrial biosphere and assciated land carbon dynamics show a clear, climate state dependent transient reaction to increasing $CO_2$ concentrations. More precisely, under conditions of the last glacial maximum, the terrestrial carbon flux balance is more sensitive to the carbon-concentration effect than under pre-industrial conditions. This is due to the lower $CO_2$ concentration in the LGM initial state that allows for a larger productivity increase under $CO_2$ concentration rise. The carbon-climate effect, in contrast, is larger under PI conditions which is caused by higher initial temperatures and larger amounts of extratropical terrestrial carbon in the PI initial state. As a consequence of this behaviour, the terrestrial feedback is stronger for LGM than PI conditions.

## 7   Code availability

The model code is publicly available after registration at www.mpimet.mpg.de/en/science/models/license.

## 8 Data availability

Simulation data are available on request from the authors.

*Author contributions.* The study was lead by M. A. who also performed the simulations and data analysis. All authors contributed to the design of the study and the manuscript.

*Competing interests.* None.

*Disclaimer.* None.

*Acknowledgements.* We would like to thank Irina Fast and the DKRZ team for their technical support and Gitta Lasslop for her careful comments on the final version of our draft. Additionally, we would like to thank our editor Christoph Heinze and the three anonymous reviewers as well as Peter Rayner for their constructive comments and suggestions. Their critical assessments helped us to significantly improve our manuscript, especially regarding the theoretical framework of the feedback analysis.

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
