# Peer review of "Earth system model simulations show different feedback strengths of the terrestrial carbon cycle under glacial and interglacial conditions"

_Earth System Dynamics, 2017_

## Author Comment (AC1) · 3 Jul 2017

Please excuse the mistaken layout for figure 8 where one axis annotation got cut off. This will be corrected with the next version upload. For the time being, please find attached the original plot.

[Figure]

[Figure]

Fig. 1.

[Figure]

---

## Referee Comment (RC1) · Anonymous Referee #1 · 20 Jul 2017

**Review of:** Earth system model simulations show different carbon cycle feedback strengths under glacial and interglacial conditions

Adloff et al.

**Overall evaluation:**

The manuscript documents an Earth system model experiment comparing terrestrial carbon cycle feedbacks under Last Glacial Maximum (LGM) and pre-industrial initial conditions. The experiments suggest that the uptake of carbon under LGM initial conditions is stronger than under pre-industrial conditions.

The manuscript is in places poorly written and generally fails to provide a convincing rational as to how the experiments increase our understanding of the Earth system. Additionally the authors seem ignorant of elementary concepts in climate science such as the definition of climate sensitivity or that the forcing from $CO_2$ is approximately a logarithmic function of concentration. Overall I recommend that the manuscript be rejected for publication in Earth system dynamics.

**General Concerns:**

**(1)** The paper is framed around exploring climate sensitivity under varying initial conditions of the climate system. However the authors appear unaware that climate sensitivity is the equilibrium change in global temperature from a doubling of atmospheric $CO_2$ concentration (IPCC AR5 Glossary). Because the forcing from $CO_2$ is approximately a logarithmic function of atmospheric $CO_2$ concentration each doubling of $CO_2$ produces approximately the same equilibrium warming. See Knutti & Hegerl (2008) for a review of equilibrium climate sensitivity.

**(2)** The experiment protocol followed in the manuscript follows the carbon cycle feedback model intercomparison project done in preparation for AR5 in with model results from CMIP5 (Arora et al. 2013). However, in numerous places in the manuscript it is stated that the experiment is following the C[4]MIP protocol. C[4]MIP used emissions driven simulations under the SRES A2 emissions scenario (Friedlingstein et al. 2006). Confusions between the two generations of model intercomparison projects demonstrated how little of the literature the authors appear to have read.

**(3)** The authors provide no sensible rational as to why conducting a pseudo-one-percent experiment at LGM initiation conditions provides any new understanding of carbon cycle feedbacks in the Earth system. From the LGM we generally want to better understand how physical and biogeochemical feedbacks combined to magnify a tiny change in the distribution of sunlight into the glacial-interglacial cycles. From the pre-industrial we are usually concerned ultimately with projecting future climate change, even in idealized experiments designed to better constrain Earth system parameters. The results of the experiments document in the manuscript are obvious a-priori given the logarithmic forcing from $CO_2$, and the reduced state of the terrestrial biosphere at the LGM.

**Specific Concerns:**

The English language is very poor in much of the manuscript. I am not systematically going to document every example but if the authors are able to salvage something publishable from these experiments please ask a native speaker read over the manuscript before re-submission.

Page 2 line 8: The sentence implies that climate sensitivity includes carbon cycle feedbacks. It does not. Climate sensitivity is measured relative to a doubling of atmospheric $CO_2$ and the atmosphere does not care where the $CO_2$ originated.

Page 2 line 29: please write out and explain the names of experiments. These abbreviations are presumably experiment codes used internally at MPI.

Page 3 line 5 and many other places: The proper term is 'radiative effect' not 'radiation effect'. In vernacular English 'radiation' alone implies ionizing radiation.

Equation 1: Why is there a colon before the equals sign?

Page 4 line 4: Using upper and lower case 'c' for different variables is confusing and prone to error. Please use more easy to distinguish symbols.

Page 5 line 11 to 14: In the 1% experiment atmospheric $CO_2$ in increased at 1% a year leading to an exponential increase in $CO_2$ concentration. Here you have used a 1% experiment based on an initial concentration of 285 ppm for both initial states. This needs to be clearly explained.

Page 6 line 12: 1) Do not abbreviate 'archipelago'. 2) The region is geographically referred to eithers as Maritime Southeast Asia, or the Malay Archipelago. The Indonesian Archipelago includes only the islands that are part of the modern nation-sate of Indonesia.

Figure 4: Why is soil water availability the only other parameter examined beyond SAT?

Page 11 line 9: Write out soil respiration instead of abbreviating to Rs.

**References:**

Arora, V. K., et al., 2013: Carbon-concentration and carbon-climate feedbacks in CMIP5 earth system models. Journal of Climate, 26 (15).

Friedlingstein, P., et al., 2006: Climate-carbon cycle feedback analysis: Results from the C4MIP model intercomparison. Journal of Climate, 19, 3337–3353.

Knutti, R. and G. C. Hegerl, 2008: The equilibrium sensitivity of the earth's temperature to radiation changes. Nature Geoscience, 1 (11), 735–743.

Planton, S., 2013: Annex III: Glossary. Climate Change 2013: The Physical Science Basis. Contribution of Working Group I to the Fifth Assessment Report of the Intergovernmental Panel on Climate Change, Cambridge University Press.

---

## Referee Comment (RC2) · Anonymous Referee #2 · 27 Jul 2017

Authors compare carbon cycle feedbacks from a pre-industrial and LGM simulation using the framework described by Friedlingstein et al. (2006) and Arora et al. (2013). Overall although the result may be somewhat obvious I still see this as a useful study as long as the underlying mechanisms are thoroughly investigated. However, the manner in which the manuscript is currently written shows that the authors haven't gained a sufficient understanding of the science as well as terminologies used in the existing literature. As such then it is clearly not of publication quality in its current form.

Main comments

My biggest concern is with the equations. On page 5 I_tot is not defined (unless I

missed it) but if I try to interpret I_tot it seems like the change in atmospheric CO2 burden. I_ext on the other hand is total cumulative emissions. If true, then the ratio between the two (equation 7) is not the feedback but rather the airborne fraction. This is not the way Friedlingstein et al. (2006) or Arora et al. (2013) described the feedback and the gain. Their feedback and the gain are calculated by comparing either simulated CO2 (in emissions-driven simulation) or diagnosed emissions (in concentration-driven simulations) from fully-coupled and biogeochemically-coupled simulations.

I am also troubled by the fact that in Figure 7 the rate of carbon uptake by land shows an abrupt slow down around CO2 concentration of 650 ppm. Figure 3c of Arora et al. (2009) shows how photosynthesis changes per unit increase in CO2 based on the standard biochemical equations for photosynthesis. Although this rate decreases, because of the saturating effect, I do not see any abrupt changes up until CO2 of 747 ppm in their figure. This abrupt behaviour in authors' model, it seems, doesn't come from the photosynthesis equations but rather something else that is implemented in the model.

Other comments

The lack of understanding of the current literature, or perhaps it's just the first language issue, is seen in several phrases used by the authors which do not appear to make any sense. These include "fertilization and radiation effect to the different vegetation distribution", "sensitivities to the fertilization and radiation effect", "when structural limits are hit", "the point of effectivity change", "physiological limits are hit more frequently", "photosynthesis exploitation of the insolation", and "tropical living conditions deteriorate".

"factorial simulations" are referred to as "factor simulations"

I have marked several other comments on the manuscript itself and am attaching a scanned version of the annotated manuscript.

References:

Arora, V. K., et al., 2013: Carbon-concentration and carbon-climate feedbacks in CMIP5 earth system models. Journal of Climate, 26 (15).

Arora, V. K. et al. The Effect of Terrestrial Photosynthesis Down Regulation on the Twentieth-Century Carbon Budget Simulated with the CCCma Earth System Model. J. Clim. 22, 6066–6088 (2009).

Friedlingstein, P., et al., 2006: Climate-carbon cycle feedback analysis: Results from the C4MIP model intercomparison. Journal of Climate, 19, 3337–3353.

Please also note the supplement to this comment:
https://www.earth-syst-dynam-discuss.net/esd-2017-67/esd-2017-67-RC2-supplement.pdf

[Figure]

**Supplement:**

[Figure]

**Earth system model simulations show different carbon cycle feedback strengths under glacial and interglacial conditions**

Markus Adloff[1,2,3], Christian H. Reick[1], and Martin Claussen[1,3]

[1]Max Planck Institute for Meteorology, Bundesstraße 53, 20146 Hamburg, Germany
[2]now: School of Geographical Sciences, University of Bristol, University Road, BS8 1SS, United Kingdom
[3]Meteorological Institute, Centrum für Erdsystemforschung und Nachhaltigkeit (CEN), Universität Hamburg, Germany
*Correspondence to:* Markus Adloff (markus.adloff@bristol.ac.uk)

*introduce with units*

**Abstract.** In Earth system model simulations we find different carbon cycle sensitivities for recent and glacial climate. This result is obtained by comparing the transient response of the terrestrial carbon cycle to a fast and strong atmospheric $CO_2$ concentration increase (roughly 1000ppm) in $C^4$MIP type simulations starting from climate conditions of the Last Glacial Maximum ("LGM") and from Pre-Industrial times ("PI"). The sensitivity $\beta$ to $CO_2$ fertilization is larger in the LGM experiment

5   during most of the simulation time: The fertilization effect leads to a terrestrial carbon gain in the LGM experiment almost twice as large as in the PI experiment. The larger fertilization effect in the LGM experiment is caused by the stronger initial $CO_2$ limitation of photosynthesis, implying a stronger potential for its release upon $CO_2$ concentration increase. In contrast, the sensitivity $\gamma$ to climate change induced by the radiation effect of rising $CO_2$ is larger in the PI experiment for most of the simulation time. Yet, climate change is less pronounced in the PI experiment, resulting in only slightly higher terrestrial

10   carbon losses than in the LGM experiment. The stronger climate sensitivity in the PI experiment results from the vastly more extratropical soil carbon under those interglacial conditions whose respiration is enhanced under climate change. Comparing the radiation and fertilization effect in a factor analysis, we find that they are almost additive, i.e. their synergy is small in the global sum of carbon changes. From this additivity, we find that the carbon cycle feedback strength is more negative in the LGM than in the PI simulations.

*introduce with units*

*unclear*

*factorial*

*unclear*

+ *don't need additivity to conclude this. You can do a straight simulation with $CO_2$ and radiative effects together. ie fully-coupled.*

**15   1   Introduction**

During the last glacial maximum (21 000 yrs before present) vegetation was not only less widespread than today but also primary productivity was smaller (Prentice and Harrison, 2009). This is the consequence of the lower $CO_2$ concentrations during those times (about 200 ppm less than today), physically via the resulting lower temperatures (greenhouse effect), and biogeochemically via the reduced photosynthetic activity ($CO_2$ fertilization effect) (Prentice and Harrison, 2009). Today $CO_2$

20   concentrations are dramatically rising and are expected to rise at least by a similar magnitude (Flato et al., 2013). One might thus hope that an analysis of the past rise will help foreseeing what to expect.   *you mean 200 ppm (by when?)*

Ideally, one could convert $CO_2$ concentration rise directly into climate and environmental changes. Accordingly, the climate community has put much effort in deriving a characteristic number for the global temperature rise induced per ppm $CO_2$ concentration rise, called "climate sensitivity". Values can be derived from paleo records and from numerical simulations

[Figure]

*→ what about future simulation that we perform*

of past climates (see e.g. PALEOSENSE (2012); Royer (2016)). Despite these intense activities the resulting values are still subject to considerable uncertainty (PALEOSENSE, 2012; Royer, 2016). But even if one had an exact number for past climates, it would not be clear whether it could be applied to the future climate development. In fact, there is evidence for a state dependence of climate sensitivity (PALEOSENSE, 2012; Woillez et al., 2011; Claussen et al., 2013). In numerical simulations,

5  this climate state dependence can be traced back to the different time scales of the forcing and feedback mechanisms (von der Heydt and Ashwin, 2016). There is also evidence that the strength of feedbacks in the climate system varies with boundary conditions (Caballero and Huber, 2013; Lunt et al., 2016)

*← do you mean initial condition*

One such feedback arises from the interaction between carbon cycle and climate. To characterize this feedback, Friedling-stein et al. (2003) introduced two carbon cycle sensitivities, both characterizing the change in stored carbon (terrestrial and/or

10  ocean) but due to different drivers: due to a change in plant available atmospheric $CO_2$ ($\beta$ sensitivity) and to a change in surface temperature ($\gamma$ sensitivity). Values have been derived from numerous Earth system simulations, particularly within the international Coupled Climate Carbon Cycle Model Intercomparison Project ($C^4$MIP)(see e.g. Friedlingstein et al. (2006); Ciais et al. (2013)). While these attempts concentrate on perturbations of the pre-industrial climate, attempts to study carbon sensitivities

*need better intro of $\gamma$ & $\beta$.*

for perturbations of past climates are rare. They all relate reconstructions of atmospheric $CO_2$ concentrations to reconstructions

15  of temperature (see Friedlingstein (2015) for a review). The resulting 'observed' sensitivity of atmospheric $CO_2$ concentration to temperature involves the combined effect of changing temperature and changing of plant available $CO_2$ and is thus neither measuring $\beta$ nor $\gamma$ as defined by Friedlingstein et al. (2003). An exception is the study by Frank et al. (2010), who considered temperature and $CO_2$ reconstructions for the last Millennium before the industrial revolution: Their estimate should be a good proxy for $\gamma$ since during this time the changes in atmospheric $CO_2$ concentration have been only a few ppm. The $\gamma$ sensitivity

*reword*

20  obtained in this way turns out to be compatible with the low end values found in the $C^4$MIP studies. *not sure what this means*

Although the contribution of the fertilization and the radiation effect to the different vegetation distribution in modern and past times have been quantified (eg. Claussen et al. (2013), Woillez et al. (2011)), there seem to be no estimates of carbon cycle sensitivities from climate simulations of the past. This makes it difficult to estimate the role carbon cycle feedbacks play for the climate state dependence of the Earth system response to rising $CO_2$ concentrations. Yet this is a critical point when

25  we want to learn from the past what future consequences of anthropogenic $CO_2$ emissions to expect. To work towards closing this gap, the present study compares the carbon cycle sensitvities of the Earth system at two different climatic conditions. In order to derive $\beta$ and $\gamma$ values for past and present we perform a set of $C^4$MIP-type simulations starting from a climate state representing the last glacial maximum and compare it with a corresponding set of $C^4$MIP-type simulations starting from a pre-industrial climate state (simulations 1pctCO2, esmFdbk1 and esmFixClim1 from $C^4$MIP). In this way we obtain $\beta$ and $\gamma$

30  values for two drastically different climates within the same model setup to evaluate their climate state dependence and the causes for this dependence. *→ reader doesn't know what these means.*

The paper is organized as follows: We begin with a detailed discussion of metrics for the analysis of carbon cycle feedbacks before applying them to the results of experiments described thereafter. Following that, we compare the different initial Earth system states in the two experiments, before analyzing the reaction to rising $CO_2$ concentration. We do this by calculating the

*no it doesn't. You are doing simulations for past. and estimate $\gamma$ and $\beta$.*

*People do it for future. There are many $C^4$MIP simulations. Explain what does this mean*

[Figure]

[Figure]

above mentioned sensitivities and the strength of the carbon cycle feedback. Finally we discuss the mechanisms underlying the differences in system behaviour.

*first time I've heard this*

**2 Disentangling the two effects of rising $CO_2$ concentrations in simulations**

5 There are two conceptionally different aspects of how rising $CO_2$ concentrations affect the terrestrial carbon cycle: A fertilization and a radiation effect. The fertilization effect leads to increased photosynthetic productivity due to more physiologically *phrase?* available $CO_2$ in the air but indirectly also to increased soil water availability because plants become more efficient in water use under increased $CO_2$ concentrations (see e.g. Chapin III et al. (2011)). The radiation effect is caused by the $CO_2$ acting as a greenhouse gas. The resulting climate change (temperature, precipitation, ...) alters the conditions for plant growth and decomposition of organic matter (litter, soils). While the strength of the fertilization effect is characterized by $\beta$, the radiation 10 effect is characterized by $\gamma$. *Units??*

To determine $\beta$ and $\gamma$ in Earth system simulations, we follow the C[4]MIP experimental design (Ciais et al., 2013, Box 6.4) for *concentration driven* simulations: Starting from a control simulation ("ctrl") performed at constant $CO_2$ concentration, two transient simulations forced by rising $CO_2$ concentrations are performed. In the first of those transient simulations ("fert") only the fertilization effect is active, which means that the rising $CO_2$ concentration is "seen" only by the photosynthesis code of the 15 model, while the radiation code constantly "sees" the $CO_2$ value of the control simulation. Conversely, in the second transient simulation ("rad") only the radiation effect is active, i.e. only the radiation code "sees" the rising $CO_2$ concentrations but not the photosynthesis model. In addition we perform a third fully coupled transient simulation ("full"), where both effects are simultaneously active. One reason for this additional simulation is to supplement our sensitivity analysis by a factor analysis *factorial* following Stein and Alpert (1993) to quantify also the synergies between the two effects. *??*

*global*

20 Our analysis focusses on changes in land carbon, denoted as $C$ in the following. For this variable the factor analysis proceeds as follows. The pure effects of fertilization and radiation are contained in the differences *phrase?*

$$
\begin{aligned}
\Delta C_{fert}(t) &:= C_{fert}(t) - C_{ctrl} \\
\Delta C_{rad}(t) &:= C_{rad}(t) - C_{ctrl}
\end{aligned}
\tag{1}
$$

where the indices at the right hand side $C$-values refer to the simulations from which the values were obtained, while the indices to the $\Delta C$-values refer to the effect considered. The time dependence $t$ appears only for the values from the transient 25 simulations, but not from the control simulations where the amount of terrestrial carbon is equilibrated in a spin up simulation. In addition the synergy is that part of the land carbon storage difference between the "full" and "ctrl" simulation that cannot be explained by an addition of the individual fertilization and radiation effects, i.e.

$$
\Delta C_{syn}(t) = (C_{full}(t) - C_{ctrl}) - (\Delta C_{fert}(t) + \Delta C_{rad}(t)).
\tag{2}
$$

*more like non-linearity*

/3

[Figure]

To obtain the $\beta$ and $\gamma$ sensitivities one must consider also differences in land temperature and atmospheric $CO_2$ concentration that develop in the transient simulations. Of particular interest are

$$
\begin{aligned}
\Delta T_{rad}(t) &:= T_{rad}(t) - T_{ctrl} \\
\Delta c(t) &:= c(t) - c_{ctrl}.
\end{aligned}
\tag{3}
$$

The concentration of atmospheric $CO_2$ is denoted here by a small "$c$" and measured in ppm $CO_2$. Since $c(t)$ is the same for the transient simulations performed here, the index refering to the type of experiment has been omitted. With these definitions the two sensitivities are defined as

$$
\begin{aligned}
\beta(t) &:= \frac{\Delta C_{fert}(t)}{\Delta c(t)} \\
\gamma(t) &:= \frac{\Delta C_{rad}(t)}{\Delta T_{rad}(t)}.
\end{aligned}
\tag{4}
$$

Finally, this analysis framework allows to quantify the strength of the carbon climate feedback. For this, in addition to $\gamma$ and $\beta$, we need to know the response of temperature to increasing $CO_2$ concentrations, expressed by the climate sensitivity $\alpha$:

$$
\alpha(t) := \frac{\Delta T_{rad}(t)}{\Delta c(t)}
\tag{5}
$$

In the considered concentration driven C$^4$MIP experiment set up, the carbon flux from the atmosphere to land carbon pools does not feed back on the atmospheric $CO_2$ concentration because the latter is prescribed. Still, the feedback strength $f$ and the carbon gain $g$ of the land carbon pools characterizing the carbon cycle feedback to rising $CO_2$ concentration can be diagnosed (Gregory et al., 2009; Arora et al., 2013). Supposing that the radiation and the fertilization effect as characterized by the sensitivities $\beta$ and $\gamma$ add up linearly, the cumulated carbon influx to the atmosphere until time $t$ consistent with the atmospheric $CO_2$ is

*[handwritten: ⤷ cumulative        Is this $I_{tot}(t)$]*

$$
I_{tot}(t) = m\Delta c(t) = I_{ext}(t) - \alpha(t)\gamma(t)\Delta c(t) - \beta(t)\Delta c(t),
\tag{6}
$$

where $I_{ext}$ is the cumulated carbon flux from external sources of emissions and $m = 2.12 \cdot 10^6$ Pg is the conversion factor from atmospheric $CO_2$ concentration to atmospheric $CO_2$ mass (Flato et al. (2013), page 471). In this diagnostic carbon balance the terms for the feedback contributions from the radiation and fertilization effects enter with a minus sign because by definition positivity of $\alpha$ and $\beta$ mean land carbon uptake and thus atmospheric carbon loss. Contributions from the ocean response to changes in $CO_2$ could be omitted in (6) since in the experiments considered here atmospheric $CO_2$ is prescribed. Following Friedlingstein et al. (2003) the feedback factor $f(t)$ and gain $g(t)$ can now be defined as

$$
I_{tot}(t) =: f(t)I_{ext}(t) =: \frac{1}{1 - g(t)}I_{ext}(t).
\tag{7}
$$

*[handwritten: → looks like airborne fraction        But $I_{ext}$ needs ocean uptake]*

By solving for $\Delta c(t)$ one obtains from (6) and the definitions (7)

$$
\begin{aligned}
f(t) &= \frac{m}{m + \alpha(t)\gamma(t) + \beta(t)} \\
g(t) &= -\frac{\alpha(t)\gamma(t) + \beta(t)}{m}
\end{aligned}
\tag{8}
$$

Note that in this framework also $\alpha$, $\beta$ and $\gamma$ are time dependent – a point that will be further discussed below.

[Figure]

**3 Experiment set up**

This paper focusses on the difference of the carbon cycle response to rising $CO_2$ concentrations in glacial and pre-industrial times. Accordingly, for each case we need a set of four simulations (ctrl, rad, fert, full) to calculate sensitivities and feedback strength. In the following, we will use the term 'experiment' to refer to one of the two cases LGM or PI. 'Simulation' will refer

5 to one of the four model runs ctrl, rad, fert or full. For the LGM case the simulations start with Last Glacial Maximum (LGM) conditions (185 ppm) and those of the other experiment start with Pre-Industrial (PI) conditions (285 ppm). We did not run the simulations for the PI experiment anew but chose to use the published CMIP5 simulations piControl, esmFdbk1, esmFixClim1 and 1pctCo2 and to set up the LGM experiment accordingly, using the same model version. The LGM simulations are initialized with restart files from an existing last glacial maximum spin-up experiment (1800 simulation years long) followed by 200 years

10 for the adaptation of dynamic vegetation (Jungclaus et al., 2014). The PI simulations are initialized with a spin-up experiment simulating climate conditions of the early 19th century over more than 3000 years (Giorgetta et al., 2012). For the transient simulations fert, rad and full, the same absolute increase in atmospheric $CO_2$ concentration is simulated in both experiments over a period of 150 years (see Fig. 1) with the full (for simulation "full") and partially coupled (for simulations "fert" and "rad") Earth system.

[Figure]

*1%, our goes to 140 years
These seem to go past
140 yrs.*

Figure 1. $CO_2$ scenarios ($\Delta c(t)$ in previous equations) as prescribed for the LGM and PI experiments: Starting from 185ppm ("Last Glacial Maximum", green line) and starting from 285ppm ("Pre-Industrial", red line).

15     The experiments are conducted with the Earth-System Model of the Max Planck Institute (MPI-ESM, compare Giorgetta (2013)). The MPI-ESM consists of the atmosphere component ECHAM6 and the ocean component MPIOM. The terrestrial processes including carbon cycle and dynamic biogeography are calculated in the land surface model JSBACH. Because atmospheric $CO_2$ concentrations are prescribed in our experiments, the oceanic and terrestrial carbon cycles are decoupled so that changes in the ocean carbon cycle are irrelevant here; nevertheless the physical ocean remains an important component

20 of the climate dynamics. JSBACH comprises the DYNVEG model for simulation of natural land cover changes (Reick et al.,

2013) and the BETHY model (Knorr, 2000) for representation of the fast processes of the biosphere, building on primary production rates that are simulated following the Farquhar model (Farquhar et al., 1980) for C3 and Collatz model (Collatz et al., 1992) for C4 photosynthesis. Vegetation is represented by eight plant functional types that differ in phenology and physiology and interact dynamically (see Brovkin et al. (2013) for an evaluation of the present implementation of dynamic

5    biogeography). Anthropogenic land cover change is not included in the experiments conducted here. Terrestrial carbon pool dynamics are calculated with CBALANCE (Reick et al., 2010), simulating temperature- and water scarcity dependent carbon fluxes between seven carbon pools with different overturning periods.

**4    Differences in the initial Earth system states**

Globally, mean near surface temperatures are 4.5 K colder in the LGM state (LGM control simulation) than in the PI state (PI

10    control simulation) but locally, temperatures differ by 20 K and more (see Fig. 2). Soil water levels are mostly higher in the LGM state, especially in the tropics and subtropics. Inland glaciers extend throughout most of North America and northern Europe and the sea level is considerably lower, leading to more landmasses, especially in the Bering Strait and the Indonesian archipel. On global scale, less area is covered by vegetation in the LGM state and dense vegetation is restricted to the tropical zone (compare Fig. 3). In the PI state, vegetation reaches far more into the extratropics and the mid latitudes are more densely

15    covered by vegetation.

[Figure]

**Figure 2.** Differences between the initial climates of the LGM and PI experiments: a) Difference between global mean near surface temperatures and b) difference between soil water availabilities. The values in the LGM initial state are substracted from the values in the PI initial state. Land areas that are covered by ice in the LGM but not PI initial state show values > 0.4 in the water availability differences.

[Figure]

[Figure]

*Show soil C as well + global totals.*

(a) LGM (b) PI

[Figure]

**Figure 3.** Potential vegetation cover in the initial states of the experiments. The degree of coverage is given in vegetation covered fractions per grid cell.

**5 Reaction of the Earth system to rising $CO_2$ concentration under different boundary conditions**

The climate system reacts differently to rising $CO_2$ concentrations for LGM and PI conditions. Fig. 4 shows changes in global mean near surface temperature and soil water availability in both experiments. Due to rising $CO_2$ concentrations, the global mean near surface temperature rises and soil water availability decreases in the global integral. Both changes are larger in the

5  LGM experiment. The temperature increases mainly due to the radiation effect of rising $CO_2$ concentrations; $CO_2$ fertilization and synergistic effects do not considerably affect global mean near surface temperature (not shown). The globally averaged soil water availability, on the contrary, rises due to increased water use efficiency in connection with the fertilization effect and decreases due to higher evapotranspiration losses under the higher temperatures as a consequence of the radiation effect. In the fully coupled run, the decrease due to climate change dominates.

10  Figure 5 shows the change of terrestrial carbon pool size due to the prescribed $CO_2$ concentration increase scenario. Overall, terrestrial carbon pool size increases in response to the rising $CO_2$ concentration in both experiments, which is due to the fertilization effect. The fertilization is stronger in the LGM than in the PI experiment. The radiation effect is negative and of similar strength in the two experiments. Synergies of both effects are small in the global integral. The last point is especially important as it shows that linear additivity of the radiation and fertilization effect can be assumed on the global scale to derive

15  the feedback strength and gain.

From Fig. 5 it becomes clear that the same absolute increase in atmospheric $CO_2$ concentration triggers different reactions of the terrestrial carbon pool in the differently initiated simulations. This is reflected in the corresponding sensitivities as shown in Fig. 6. There, the sensitivity values for the LGM and PI experiments are shown as a function of simulation time. In the following, the sensitivities to the radiation and fertilization effects will be studied one after the other, before discussing the

20  combined feedback strength.

*of land C*

[Figure]

[Figure]

**Figure 4.** Climatic changes in the fully coupled run (continuous lines) and radiatively coupled run (dashed lines) due to rising $CO_2$ concentrations in the LGM experiment (red) and PI experiment (black). a) shows the globally averaged change in near surface temperature and b) in soil water availability.

[Figure]

**Figure 5.** Change in terrestrial carbon storage [PgC] in the fully coupled simulation (black curve) and split into factors (coloured curves) as deviation from the control simulation (a) in the LGM experiment and (b) in the PI experiment.

**5.1 The fertilization effect**

In both experiments, $\beta$ increases in the beginning of the experiment. But already for simulation times larger than 30 to 40 years, the increase slows down. Arora et al. (2013) attribute this behaviour to the difference in response time of primary production and biomass decomposition. While productivity increases almost instantaneously with increasing physiologically available
5  $CO_2$, biomass decomposition remains initially unchanged and only increases when, in consequence of the higher productivity, after a temporal delay more biomass is transferred to litter and soil carbon pools. Additionally - and this is found to be the main effect in the present study - the fertilization effect becomes less effective at high productivity levels because carbon density of living vegetation is reaching upper limits. In fact, the amount of carbon allocatable to biomass carbon pools is restrained in JSBACH to account for a down regulation of carbon allocation when structural limits are hit. During the entire simulation

[Figure]

[Figure]

[Figure]

**Figure 7.** Dependence of gross assimilation per m$^2$ leaf on air $CO_2$ concentration according to the implemented photosynthesis model (Farquhar et al., 1980) for C3 plant physiology at 20°C leaf temperature. Abbreviations stand for individual vegetation types: TET for tropical evergreen trees, TDT for tropical deciduous trees, EET for extratropical evergreen trees, EDT for extratropical deciduous trees, RGS for raingreen shrubs, DCS for deciduous shrubs, C3G for C3 grasses and C4G for C4 grasses.

although the final $CO_2$ concentration is larger in the PI experiment, the fertilization effect is larger in the LGM experiment. The stronger $CO_2$ fertilization in the LGM experiment is mostly due to the strong reaction of tropical vegetation. In the extratropics, the fertilization effect is stronger in the PI experiment but still does not reach the tropical production rate increases of the LGM experiment.

**5.2 The radiation effect**

$\gamma$ grows increasingly negative in both experiments (see Fig. 6). It is increasingly larger in absolute value in the PI experiment than in the LGM experiment. Although $\gamma$ differs clearly in the two experiments, the overall terrestrial carbon pool changes in the radiatively coupled simulations are almost similar (compare Fig. 5). The reason for this is that also the climate sensitivity $\alpha$ varies between the two experiments. $\alpha$ is larger in the LGM experiment over the entire simulation time. The higher climate sensitivity and the lower carbon cycle sensitivity $\gamma$ partially compensate differences between the PI and LGM cases as is seen from Fig. 6 (d) where the product $\alpha\gamma$ has been plotted; it is this combination of sensitivities that describes the total radiative effect on carbon losses (compare equation (6)). Thereby the radiation effect on land carbon storage differs much less between the LGM and PI case than the fertilization effect discussed above.

To understand the processes behind the different $\gamma$ sensitivity in the two experiments, it is useful to analyze first how climate change induces carbon losses differently in the tropics and extratropics. Table 1 lists the change per degree temperature change in soil respiration $\Delta Rs$ relative to the one in net primary productivity $\Delta NPP$ separately for tropics and extratropics

[Figure]

[Figure]

**Figure 6.** Sensitivities $\beta$ and $\gamma$ to the fertilization and radiation effect (respectively) of rising $CO_2$ concentrations and climate sensitivity $\alpha$ in the LGM (blue) and the PI experiment (red).

time, $\beta$ is larger in the LGM experiment than in the PI experiment. This is caused by the lower initial $CO_2$ concentration. In both initial states, photosynthesis is carboxylation rate limited. In other words, the initial atmospheric $CO_2$ concentrations are too low as to allow for maximum photosynthetic exploitation of the insolation. *Phrase?* This initial $CO_2$ limitation is lifted by the increasing $CO_2$ concentration and leads to increasing primary productivity that allows for vegetation cover extension and

5   increasing terrestrial carbon pool sizes. As long as the $CO_2$ availability stays to be the main limitation for productivity, the fertilization effect of rising $CO_2$ concentration leads to large increases in productivity. In our experiments, the simulated $CO_2$ concentration rise is however large enough to reach a point where insolation becomes more limiting to productivity than the $CO_2$ availability. The transition is clearly visible in Fig. 7, where the modeled dependence of primary production rate on $CO_2$ concentration is shown for the eight vegetation types present in our simulations.

*seems like an unrealistic model feature*

10    From that transition point on, the effectivity of $CO_2$ fertilization is reduced. The prescribed $CO_2$ forcing is such that the difference in $CO_2$ concentration between the two experiments remains the same throughout the simulation time. However, because the $CO_2$ concentration is raised beyond the point of effectivity change, *Phrase?* the stronger initial $CO_2$ limitation of the LGM experiment is more important for the fertilization effect than the higher final concentration in the PI experiment. Therefore,

*not clear what is meant here*

[Figure]

in the two 'rad' simulations. In both simulations this ratio is smaller than one in the tropics (more land carbon input change ↑ na photosynthesis
than output change) *na respiration* but larger than one in the extratropics (more land carbon output change than input change), indicating
a very different functioning of the carbon cycle under climate change in these two regions. Considering first the tropics,
net primary productivity and soil respiration decrease (see table), indicating that living conditions deteriorate here. This has

5   two reasons: First, its getting drier so that plant productivity and also soil decomposition are reduced. Second, the already
    hot tropical climate is getting even hotter during the simulations so that physiological limits are hit more frequently thereby *phrase ?*
    deteriorating plant productivity by damaging the photosynthetic apparatus (implemented as 'heat inhibition' in MPI-ESM). But
    the reduction in NPP is much larger than the reduction in soil respiration. Hence in the tropics land carbon losses are mostly
    driven by reduced plant productivity. In the extratropics the situation is different: rising values of NPP and Rs (see table) are

10  well understandable because under the warming climate physiological processes speed up there. But since ultimately Rs is fed
    from NPP, the considerably larger increase in Rs cannot be a result of the enhanced carbon input. Instead, it results from the
    enhanced decomposition of soil carbon that had accumulated in those vast cold boreal areas already in the control simulation.
    Hence in the extratropics land carbon losses are mostly driven by enhanced soil respiration of 'old' carbon.

    Having identified the major drivers for carbon losses in the tropics and extratropics, one can now understand why the

15  temperature sensitivity $\gamma$ is larger in the PI than in the LGM simulation. In the tropics plant productivity reduction is the major
    driver, and productivity reacts more sensitive in the PI than the LGM simulation (see table 1) because tropical living conditions *phrases?*
    deteriorate from already initially drier and hotter conditions. And in the extratropics enhancement of soil respiration was found
    to be the major driver, and soil respiration reacts more sensitive in the PI than in the LGM simulation (see table 1) because of
    the vegetation extending much farther north under the warmer conditions and in absence of ice sheets going along with vastly

20  more extratropical 'old' soil carbon getting respired. Hence both in the tropics and in the extratropics the land carbon cycle
    is more sensitive to climate change in the PI case. Furthermore, table 1 shows that the larger sensitivity of the extratropics
    dominates the one in the tropics.   *I don't see this*

Table 1. Temperature sensitivity of net primary productivity NPP and soil respiration Rs due to the radiative effect. Sensitivities are computed
from changes $\Delta$NPP and $\Delta$Rs per temperature change $\Delta$T. Additionally, the relative change of soil respiration (soil respiration change
divided by NPP change) is given in the last row. The change of the carbon fluxes over the entire simulation time (end values minus start
values), integrated over the Earth's surface and all vegetation types is devided by the regional temperature change over the same period.
'Tropics' refers here to the latitudinal belt between 30° South and 30° North and 'extratropics' to the remaining part of the globe. Here,
$\Delta$NPP and $\Delta$Rs are considered positive for plant carbon uptake and soil carbon loss, respectively.

| sensitivity [PgC/K] | tropics | | extratropics | |
|---|---|---|---|---|
| | LGM | PI | LGM | PI |
| $\Delta$NPP/$\Delta$T | -134.6 | -151.2 | 10.8 | 28.6 |
| $\Delta$Rs/$\Delta$T | -55.9 | -49.7 | 17.1 | 48.2 |
| $\Delta$Rs/$\Delta$NPP | 0.42 | 0.33 | 1.59 | 1.69 |

[Figure]

[Figure]

**5.3 Feedback strength and atmospheric carbon gain**

[Figure]

[Figure]

**Figure 8.** Feedback strength $f$ and atmospheric carbon gain factor $g$ for the overall feedback of the carbon cycle to the imposed $CO_2$ forcing in the LGM and the PI experiment.

Due to the the radiation and the fertilization effect, the reacting terrestrial carbon pool produces a feedback to the initial $CO_2$ forcing. The strength of this terrestrial carbon cycle feedback is lower than 1 (see figure 8 (a)), showing that the total carbon cycle feedback to rising $CO_2$ concentrations dampens the effect of the forcing so that less carbon is left in the atmosphere than

5   injected by the forcing. Accordingly, the feedback is negative, as also visible from the gain factor (figure 8 (b)). This effect is stronger in the LGM experiment, especially towards the end of the simulations, due to the larger value of $\beta$. The stronger negative carbon cycle feedback in the LGM experiment would, in case of a prescribed amount of carbon input, lead to a reduced increase in atmospheric $CO_2$ concentration compared to the PI experiment. From the more negative LGM gain factor it is also understandable that despite identical concentration scenarios in the two experiments, the absolute mass of carbon introduced

10  to the system by the end of the simulation is larger in the LGM experiment. The recovery of $f$ and $g$ towards the end of the simulation is mostly a consequence of the recovery of $\beta$ (compare figure 6 (a)) for the reasons explained in section 5.1.

**6   Discussion and Conclusion**

Results from the different phases of $C^4MIP$ demonstrate that $\beta$ and $\gamma$ vary largely with the employed Earth system model (see Arora et al. (2013) and the third data column in table 2). The $\gamma$ values of the LGM and the PI experiment obtained in the

15  present study lie inside the inter-model range for the preindustrial $\gamma$ whereas the glacial $\beta$ is larger than any preindustrial $\beta$ from the inter-model comparison. It is important to mention that there is a difference in the CMIP values for the MPI-ESM and those calculated from the PI experiment in the present study. The latter consider only natural vegetation types to improve the comparability of the anthropogenically unperturbed LGM and the anthropogenically perturbed PI states of the biosphere. In contrast, anthropogenic bioms are included in the calculations of the CMIP values, which lead to larger absolute values of $\beta$

20  and $\gamma$ than with natural vegetation only. Additionally, a different time averaging is applied.

[Figure]

Table 2. $\beta$ and $\gamma$ sensitivities at the end of the LGM and PI experiments and their inter-model range according to Arora et al. (2013) for the PI experiment, considering only models without nitrogen cycle.

| sensitivity value | LGM exp. | PI exp. | Arora et al. (2013) |
|---|---|---|---|
| $\beta$ [$PgC/ppm$] | 2.07 | 1.33 | 0.74 – 1.46 |
| $\gamma$ [$PgC/K$] | -44.0 | -74.4 | -30.1 – -88.6 |

explain in simple language

While the difference in sensitivities between the LGM and the PI experiments can be traced back to different initial conditions in these experiments, there is still a strong dependence on the forcing pathway and its absolute amplitude. For example, the

5 difference in $\beta$ depends largely depends on whether the forcing is strong enough to produce a switch from carboxylation rate limited to electron transport limited assimilation. Additionally bioclimatic limits of vegetation, maximum productivity rates, the choice of the wilting point and maximum carbon pool sizes introduce transition points to the system that shape the behaviour of terrestrial carbon pools, which, in consequence, will show different reactions to forcings of different amplitudes. From the employed set up and sensitivity measures, it is therefore not possible to derive and compare equilibrium sensitivities which

reword

10 should be independent of the forcing scenario and could ideally be understood as a system property to characterise an Earth system state as such. Additionally, the absolute sensitivity and feedback values – including those found in the present study – must be considered with care since not yet all adaptational strategies of vegetation to changing climate and $CO_2$ concentration are known (e.g. Christmas et al. (2015)) and implemented in numerical models. But even if one is sceptical about the realism of the numbers obtained for the LGM and PI sensitivities, the present study has demonstrated that they can be used to understand

15 why the Earth system may react differently to rising $CO_2$ concentrations under LGM and PI conditions. In the two experiments, the terrestrial biosphere and carbon pools react differently to the same absolute increase in atmospheric $CO_2$ concentration. It can therefore be concluded that there is a climate state dependence in the transient reaction of the terrestrial carbon cycle to increasing $CO_2$ concentrations in these experiments. More precisely, for LGM conditions, the carbon flux balance is more sensitive to the fertilization effect than in PI conditions. This is due to a more severe $CO_2$ limitation of primary productivity

20 in the LGM initial state that provides more potential for relaxation. The sensitivity to the radiation effect, in contrast, is larger under PI conditions which is caused by higher initial temperatures and larger extratropical terrestrial carbon pools in the PI initial state.

phrase?

**7 Code availability**

The model code is publicly available after registration at www.mpimet.mpg.de/en/science/models/license.

[Figure]

**8 Data availability**

Simulation data are available on request from the authors.

*Author contributions.* The study was lead by M. A. who also performed the simulations and data analysis. All authors contributed to the design of the study and the manuscript.

5   *Competing interests.* None.

*Disclaimer.* None.

*Acknowledgements.* We would like to thank Irina Fast and the DKRZ team for their technical support and Gitta Lasslop for her careful comments on the final version of our draft.

---

## Author Comment (AC2) · 28 Aug 2017

We thank the reviewer for his/her comments on the submitted paper. We seem to significantly disagree on the rationale behind our study and hope for clarification in this regard to make the discussion more constructive. Language and writing style are crucial in any publication and we are very concerned to read that language made it difficult for the reviewer to understand the submitted paper. We will review our paper in this regard, for which the detailed reviewer's comments will be helpful, and make sure to collaborate with native speakers as suggested by the reviewer. In the following, we address the three main content-related comments individually.

[Figure]

ad (1): Our paper is NOT framed around climate sensitivity, but around carbon cycle feedbacks. Indeed we talk also about climate sensitivity but from the experiment design it should be clear that this is not the equilibrium climate sensitivity defined by the IPCC and we do not employ this term in our paper. In fact, what we are looking at are transient sensitivities, of the climate system as well as of the terrestrial carbon cycle specifically. Transient sensitivities are well introduced in the climate community, but the term transient climate sensitivity is not used by the IPCC – the latter uses instead the term 'transient climate response' (IPCC AR5 Glossary). We understand from the reviewer's comment that we should not shorten 'transient climate sensitivity' to 'climate sensitivity' to prevent confusions. We also want to point out that in the definition of climate sensitivity in the context of carbon cycle feedback studies (see Friedlingstein et al., 2003 eq. 3, Friedlingstein et al., 2006 and Arora et al., 2013 p. 5293) a linear dependence of the sensitivity on CO2 is assumed.

ad (2): The essence of the C4MIP protocol is in our opinion not whether simulations are concentration or emission driven, but the definition of sensitivities on the basis of three differently coupled transient simulations. This is also the IPCC view (Ciais et al., 2013, Box 6.4): in the framework of CMIP5 the C4MIP project has performed concentration AND emission driven simulations to derive the respective sensitivities. Moreover, we make very clear in the paper that we consider concentration driven simulations, e.g. on page 3, lines 11-12: "we follow the C4MIP experimental design (Ciais et al., 2013, Box 6.4) for concentration driven simulations", where we even emphasized this point by using italics. Concerning the reviewer's conclusion that our presentation shows "how little of the literature the authors appear to have read" we want to remark that such a broad statement is rather offending: It would be appropriate if the reviewer would point out more clearly and explicitly if he/she feels that a particular reference is missing or has not been sufficiently acknowledged.

ad (3): We disagree with the reviewer that LGM studies are only good for "better un-derstand(ing) how physical and biogeochemical feedbacks combine to magnify a tiny

change in the distribution of sunlight into the glacial-interglacial cycles". Besides the fact that a better understanding of the transient sensitivities of the terrestrial carbon cycle could improve our understanding of the Earth system's reaction to external forcings, we think that also other questions are of interest, namely to what extent feedbacks as quantified in the C4MIP way are different with different background climates. On the whole, we don't understand the basis on which the reviewer suggests where our interest should be in the first place. To get a more constructive feedback it would help us if the reviewer could explain why our research question is not properly stated or even not worth it studying.

From the comments of the second reviewer we understand that we caused confusion by refering to Friedlingstein et al. 2003 in our methods section. Indeed, the concept of carbon cycle sensitivities and the carbon cycle feedback originated from Friedlingstein's work but Gregory et al. 2009 altered the definition particularly of the carbon cycle feedback to include the overall feedback from the carbon cycle and not only it's radiative part. Our calculations follow Gregory's altered definition to quantify the overall feedback. In the re-submitted paper, we will make sure to avoid this confusion.

Arora, V. K., et al.: The effect of terrestrial photosynthesis down regulation on the twentieth-century carbon budget simulated with the CCCma Earth System Model. Journal of Climate, 22.22, pp. 6066-6088, 2009. Ciais, P., et al.: Carbon and Other Biogeochemical Cycles, in: Climate Change 2013: The Physical Science Basis. Contribution of Working Group I to the Fifth Assessment Report of the Intergovernmental Panel on Climate Change, edited by Stocker, T., et al., pp. 465–570, Cambridge University Press, Cambridge, UK, and New York, NY, USA, 2013. Friedlingstein, P., Dufresne, J.-L., and Cox, P. M.: How positive is the feedback between climate change and the carbon cycle?,Tellus, 55B, pp. 692–700, 2003 Friedlingstein, P., Cox, P., Betts, R., Bopp, L., von Bloh, W., Brovkin, V., Cadule, P., Doney, S., Eby, M., Fung, I., Bala, G., John, J., Jones, C. D., Joos, F., Kato, T., Kawamiya, M., Knorr, W., Lindsay, K., Metthews, H. D., Raddatz, T., Rayner, P., Reick, C., Roeckner, E., Schnitzler, K.-G.,
Schnur, R., Strassmann, K., Weaver, A. J., Yoshikawa, C., and Zeng, N.: Climate-Carbon Cycle Feedback Analysis: Results from the C4MIP Model Intercomparison, Journal of Climate, pp. 3337–3353, 2006.

---

## Author Comment (AC3) · 28 Aug 2017

Response to the review comments by Anonymous Reviewer 2:

We thank the reviewer for the comments on the submitted study and especially for the very detailed comments in the scanned paper that will be useful in preparing our manuscript for resubmission.

The main concerns of the reviewer are related to our equations in chapter 2. In the introduction of our equations, we follow Gregory et al. 2009. Just as the reviewer mentioned, $I\_tot$ in equation 6 of our paper is "the cumulated carbon influx to the at-

mosphere until time t consistent with the atmospheric CO2" (see our remark above eq. 6). I_ext is the cumulated emissions from external sources, as pointed out immediately underneath the equation. One can think of the latter one being the amount of carbon emissions necessary to get the change of atmospheric carbon content that is prescribed in our concentration driven experiments. The difference between I_tot and I_ext is just the amount of carbon that is added/removed via feedbacks. As correctly realized by the reviewer, the quotient I_tot/I_ext is indeed the airborne fraction. But it is as well the feedback strength (Gregory et al. 2009, p. 5238) so that the reviewer's and our interpretation of this quotient are correct. When revising our paper we will point out these alternative interpretations and will take the opportunity to compare with published estimates of the airborne fraction in MPI-ESM and other simulations. – We thank the reviewer for this hint.

From the reviewer's comments we realize that we produced some confusion by incorrectly pointing to Friedlingstein et al 2003 when introducing the feedback factor and gain in the context of eq. 7: our feedback factor and gain are those of Gregory et al. 2009 (the latter named there gC) and not those of Friedlingstein et al. 2003 (Friedlingstein g is called gCC in Gregory et al. 2009) as correctly spotted by the reviewer. The difference is that gC characterizes the feedback induced by the additional radiative forcing and CO2 fertilization of the CO2 emissions, while the Friedlingstein g characterizes only the feedback induced by the radiative forcing. Since we are interested in the system as a whole, we prefer to use gC. When resubmitting we will take care to prevent this confusion.

Another concern expressed by the reviewer is that the sharp transition of the dependence of assimilation rate on atmospheric CO2 concentration shown in figure 7 of our submitted paper cannot be recognized in figure 3c in Arora et al. 2009. This observation does not represent a contradiction between the two studies: Figure 7 in our paper shows the transition between two 'modes' of photosynthetic assimilation as a function of CO2 concentration. In each 'mode' a different subprocess is limiting the

assimilation rate as a whole: carboyxlation or electron transport. In order to assess which of the two limitations presents a larger constraint to the assimilation rate, our model calculates carboxylation rate and electron transport rate and calculates the resulting assimilation rate based on the smaller of the two. This is the same in the CTEM model used by Arora et al. 2009 – see their equations (4) and (5) that describe the two modes. The difference between our figure 7 and figure 3c in Arora et al. 2009 is thus that we display how the assimilation rate resulting simultaneously from both limitations depends on $CO_2$, whereas Arora et al. 2009 show the $CO_2$ dependence of carboxylation and electron transport rate individually. We included figure 7 into our study to point out that assimilation rates are considerably less sensitive to rising $CO_2$ concentration after $CO_2$ has reached the transition point from carboxylation to electron transport rate limitation. With the preindustrial $CO_2$ concentration being closer to the transition point than the glacial $CO_2$ concentration, the regime where the assimilation rate saturates by electron transport limitation is reached much earlier during the scenario starting from pre-industrial conditions, which is the main cause for the smaller sensitivity to rising $CO_2$ concentrations in this case.

As with the first review comment, we take the remarks on language and style very seriously. We will make sure that the next manuscript version is of satisfactory style for native speakers. Here, we intend to clarify exemplarily the criticized formulations explicitly cited by the reviewer:

"fertilization and radiation effect to the different vegetation distribution" -> The following formulation (in the full sentence) might be clearer: "the contribution of the fertilization and radiation effect of different ambient $CO_2$ concentrations to the difference in vegetation distribution during glacial and preindustrial times"

"sensitivity to the fertilization or radiation effect"-> We use this expression because some carbon fluxes are only sensitive to one of the two effects of rising $CO_2$ concentrations. For example, autotrophic respiration is directly affected only by the radiation effect.

"when structural limits are hit" -> The word 'structural' refers here to the way carbon pools are set up in the model. Some carbon pools in our model have upper limits of carbon they can contain. When these limits are reached, carbon allocation to these pools stops. We could reformulate this to: "when the upper limit of carbon assumed to be allocatable to these pools is reached"

"physiological limits are hit more frequently" -> This can be reformulated to: "physiological limitations are reached more frequently"

"photosynthetic exploitation of the insolation" -> We use this formulation to speak about the amount of radiative energy in PAR that can be used for photosynthesis under CO2 availability limitation.

"tropical living conditions deteriorate" -> This could be reformulated as: "growth conditions in the tropics deteriorate"

One reason for the linguistic confusion might be that we used a terminology for feedbacks and experiment simulations that is different from previous studies. It is important to mention that there is no agreed, commonly used terminology in the literature yet but we will make sure to introduce our terminology more carefully and to link it to the different terms used in other studies.

Arora, V. K., et al.: The effect of terrestrial photosynthesis down regulation on the twentieth-century carbon budget simulated with the CCCma Earth System Model. Journal of Climate, 22.22 (2009): 6066-6088. Friedlingstein, P., Dufresne, J.-L., and Cox, P. M.: How positive is the feedback between climate change and the carbon cycle?,Tellus, 55B, 692–700, 2003 Gregory, J. M., Jones, C. C., Cadule, P., and Friedlingstein, P.: Quantifying Carbon Cycle Feedbacks, Journal of Climate, 22, 5232–5250, 2009.

---

## Author Response (AR1)

Dear Dr. Heinze,

Thank you for the opportunity to improve our paper and resubmit it to Earth System Dynamics! Your and the reviewer's comments identified important shortcomings of our paper. In response to the comments we have re-written large parts of the manuscript and hope that it is now better understandable and more concise. In particular we followed your suggestion to stick closer to the terminology widely used in the relevant literature so that we now use the more common terms 'carbon-concentration effect' and 'carbon-climate effect' instead of our namings 'radiation effect' and 'fertilization effect'.

Concerning our loose usage of the term 'climate sensitivity', we recognized that the large room this topic took especially in our introduction was inappropriate, and may partly explain the severe criticism of both reviewers of our use of this term although this is only a side topic in our study. We now talk when needed more precisely of 'temperature sensitivity', which is the only 'climate sensitivity' relevant in our context. We hope our readers will find our revised introduction now more focused on the core topics of our study and that climate sensitivity has now the appropriate weight in our presentation.

We thank you for pointing us to Roe (2009). In the revised manuscript, the mathematical framework does now comply with Roe (2009). Although feedback strength and gain have been used differently in previous carbon cycle analyses (e.g. Gregory et al 2009 and Arora et al 2013), we understand the importance of using terms according to their mechanistic meaning. Accordingly, we now follow the original terminology from the engineering literature as recommended by Roe (2009).

We particularly thank you and your discussion partners including reviewer #2 for pointing us to a serious shortcoming in our presentation of the necessary feedback formalism. It is indeed true that in our simulations land and ocean carbon cycle are decoupled because we use prescribed atmospheric CO2. But thinking that therefore we could silently ignore the ocean in the diagnostic global carbon budget (our former eq. (6)) was indeed misleading, since in this way the interpretation of the feedback factor computed in our study as a measure for the terrestrial contribution to the overall feedback (which includes also the contribution from the ocean) was obscured. We hope that with the revised presentation of our methodology, we can convince you and the reviewers that even though our study concentrates only on the terrestrial carbon cycle, our study makes a reasonable contribution towards the understanding of differences in climate – carbon cycle feedbacks between pre-industrial times and the during the last glacial maximum.

We thank you for pointing out that the additivity of the carbon-concentration and carbon-climate effect is a model specific feature. We state this clearly in the revised manuscript and refer as you suggested to Gregory (2009).

As you suggest, we discuss now in the revised manuscript the difference between Delta T\_rad and Delta T\_full, although it is smaller in the MPI-ESM model then in others. In that context, we now also point out that this difference is smaller in the glacial than the interglacial experiment.

Concerns about unclear and misleading use of language have been raised by all reviewers. During the preparation of the new version of our manuscript, we have reviewed the wording, particularly in the instances raised by the reviewers and took help of native speakers.

The new version of our manuscript is hopefully more coherent both in itself and with previous studies. The new introduction and methods chapter provide a clearer motivation for our study and the scientific background and a more rigorous derivation of our analytical framework, respectively.

Additionally, thanks to the intervention of you and the reviewers, we are confident that the new version is easier to read and understand.

We very much appreciate your balanced judgement, particularly in view of the very sceptic reviews.

With best regards, Markus Adloff, Christian Reick and Martin Claussen We thank Reviewer 1 for her/his constructive comments. In the following, we respond to her/his comments (cited by using grey, italic fonts) and explain how we addressed the points raised in the preparation of the revised version of our manuscript.

**Overall evaluation:**

The manuscript documents an Earth system model experiment comparing terrestrial carbon cycle feedbacks under Last Glacial Maximum (LGM) and pre-industrial initial conditions. The experiments suggest that the uptake of carbon under LGM initial conditions is stronger than under pre-industrial conditions.

Our experiments suggest that the terrestrial carbon cycle reacts more sensitively to rising CO2 concentrations under LGM than under pre-industrial conditions. Moreover, we also quantified this different feedback strength and investigated the underlying processes. Obviously we failed to convey our main results to the reader. Therefore we have largely re-written the paper and hope that our main results are now clearly recognized.

The manuscript is in places poorly written and generally fails to provide a convincing rational as to how the experiments increase our understanding of the Earth system.

**We now largely reformulated the paper and took advice from native speakers.**

Additionally the authors seem ignorant of elementary concepts in climate science such as the definition of climate sensitivity or that the forcing from CO 2 is approximately a logarithmic function of concentration.

Concerning climate sensitivity we comment below. Concernig the logarithmic dependence of the forcing on  $CO_2$  concentration: Yes, we are aware of this, and indeed this would be a problem if one would understand the Friedlingstein feedback formalism that we employ as derived from a Taylor series expansion in  $CO_2$  and temperature. But this would be a misunderstanding: The Friedlingstein sensitivities are time dependent and therefore implicitly account for this logarithmic dependency.

*Overall I recommend that the manuscript be rejected for publication in Earth system dynamics.*

**We are sorry to read this.**

**General concerns:**

(1) The paper is framed around exploring climate sensitivity under varying initial conditions of the climate system. However the authors appear unaware that climate sensitivity is the equilibrium change in global temperature from a doubling of atmospheric  $CO_2$  concentration (IPCC AR5 Glossary). Because the forcing from  $CO_2$  is approximately a logarithmic function of atmospheric  $CO_2$  concentration each doubling of  $CO_2$  produces approximately the same equilibrium warming. See Knutti & Hegerl (2008) for a review of equilibrium climate sensitivity.

Our paper is NOT framed around climate sensitivity, but around carbon cycle feedbacks. Indeed we talk also about climate sensitivity but from the experiment design it should be clear that this is not the equilibrium climate sensitivity defined by the IPCC, and we do not employ this term in our paper. In fact, what we are looking at are transient sensitivities, of the climate system as well as of

the terrestrial carbon cycle specifically. To prevent confusion, we now use the term 'temperature sensitivity' in the new manuscript. We also want to point out that in the definition of temperature sensitivity in the context of carbon cycle feedback studies (see Friedlingstein et al., 2003 eq. 3, Friedlingstein et al., 2006 and Arora et al., 2013 p. 5293) a linear dependence of the sensitivity on CO2 is assumed. Additionally, we do not discuss climate sensitivity any more in the introduction since it is not central to our study and obviously produced confusion.

(2) The experiment protocol followed in the manuscript follows the carbon cycle feedback model intercomparison project done in preparation for AR5 in with model results from CMIP5 (Arora et al. 2013). However, in numerous places in the manuscript it is stated that the experiment is following the C4MIP protocol. C4MIP used emissions driven simulations under the SRES A2 emissions scenario (Friedlingstein et al. 2006). Confusions between the two generations of model intercomparison projects demonstrated how little of the literature the authors appear to have read.

The essence of the C4MIP protocol is, in our opinion, not whether simulations are concentration or emission driven, but the definition of sensitivities on the basis of three differently coupled transient simulations. This is also the IPCC view (Ciais et al., 2013, Box 6.4): in the framework of CMIP5 the C4MIP project has performed concentration AND emission driven simulations to derive the respective sensitivities. Moreover, we make very clear in the paper that we consider concentration driven simulations, e.g. on page 3, lines 20-21: "we follow the C4MIP experimental design (Ciais et al., 2013, Box 6.4) in the variant of *concentration driven* simulations", where we emphasized this point by using italics.

(3) The authors provide no sensible rational as to why conducting a pseudo-one-percent experiment at LGM initiation conditions provides any new understanding of carbon cycle feedbacks in the Earth system. From the LGM we generally want to better understand how physical and biogeochemical feedbacks combined to magnify a tiny change in the distribution of sunlight into the glacial-interglacial cycles. From the pre-industrial we are usually concerned ultimately with projecting future climate change, even in idealized experiments designed to better constrain Earth system parameters. The results of the experiments document in the manuscript are obvious a-priori given the logarithmic forcing from  $CO_2$ , and the reduced state of the terrestrial biosphere at the LGM.

Besides the fact that a better understanding of the transient sensitivities of the terrestrial carbon cycle could improve our understanding of the Earth system's reaction to external forcings, we think that also other questions than mentioned by the reviewer are of interest, namely to what extent feedbacks as quantified following the C4MIP protocol are different under different background climates. Our research indicates that the terrestrial carbon cycle is more sensitive to the carbon-concentration effect but less sensitive to the carbon-climate effect under glacial than under interglacial conditions. We are also able to identify particular aspects of the background Earth system state that cause these differences. The results did not seem obvious to us a priori and have not been shown by other published studies.

**Specific Concerns:**

The English language is very poor in much of the manuscript. I am not systematically going to document every example but if the authors are able to salvage something publishable from these experiments please ask a native speaker read over the manuscript before resubmission.

We reviewed our writing style in terms of word use, grammar and structure during the preparation of the new manuscript, based on advice from native speakers.

Page 2 line 8: The sentence implies that climate sensitivity includes carbon cycle feedbacks. It does not. Climate sensitivity is measured relative to a doubling of atmospheric  $CO_2$  and the atmosphere does not care where the  $CO_2$  originated.

**We decided to restructure the introduction and to skip the discussion of climate sensitivity as the previous version proved to be misleading.**

*Page 2 line 29: please write out and explain the names of experiments. These abbreviations are presumably experiment codes used internally at MPI.*

The experiment names are the official names used in the CMIP5 protocol. We use them here explicitely so that the reader can find the exact experiment procedure and results in the CMIP5 documents and archives.

*Page 3 line 5 and many other places: The proper term is 'radiative effect' not 'radiation effect'. In vernacular English 'radiation' alone implies ionizing radiation.*

In the previous version, we had introduced the terms 'radiation effect' and 'fertilization effect' to differentiate between the effect rising CO2 concentrations have on the radiative balance of the Earth system and on the productivity of land plants. The terms 'carbon-concentration' and 'carbon-climate' effect used in previous studies do not make this difference clear because both, greenhouse effect and enhanced productivity are due to higher CO2 concentrations (so could be called 'carbon-concentration' effects). However, our wordings seem to have been more confusing than helpful for the reviewers. Therefore, we switched to the terminology used in previous studies.

**Equation 1: Why is there a colon before the equals sign?**

This is a standard notation in mathematics to indicate the term on the side of the colon is defined by the term at the other side of the equality sign. Thereby one distinguishes definitions from conclusions.

*Page 4 line 4: Using upper and lower case 'c' for different variables is confusing and prone to error. Please use more easy to distinguish symbols.*

**We agree that using 'c' and 'C' in the same text can easily confuse the reader and, hence, changed symbols in our new manuscript.**

Page 5 line 11 to 14: In the 1% experiment atmospheric  $CO_2$  in increased at 1% a year leading to an exponential increase in  $CO_2$  concentration. Here you have used a 1% experiment based on an initial concentration of 285 ppm for both initial states. This needs to be clearly explained.

When introducing our experiments we clearly state that the same absolute amount of  $CO_2$  increase is prescribed in both experiments instead of saying that  $CO_2$  concentrations increase by 1% per year. Additionally, we show the concentration changes in Fig. 1.

Page 6 line 12: 1) Do not abbreviate 'archipelago'. 2) The region is geographically referred to eithers as Maritime Southeast Asia, or the Malay Archipelago. The Indonesian Archipelago includes only the islands that are part of the modern nation-sate of Indonesia.

We agree with the reviewer that we used a wrong geographic term here and, hence, changed it in the manuscript.

**Figure 4:* Why is soil water availability the only other parameter examined beyond SAT?**

Near surface air temperature and plant water availability are the most important climatic variables that influence terrestrial carbon fluxes in the employed model. In the new version of the manuscript version, we now motivate the analysis of plant water availability in this way.

**Page 11 line 9: Write out soil respiration instead of abbreviating to Rs.**

In the new version of the manuscript, we follow the reviewer's suggestion to write out soil respiration in the text instead of abbreviating it. We hope that doing so improves the readability of the text.

**References:**

Arora, V. K., et al.: The effect of terrestrial photosynthesis down regulation on the twentieth-century carbon budget simulated with the CCCma Earth System Model. Journal of Climate, 22.22 (2009): 6066-6088. Ciais, P., et al.: Carbon and Other Biogeochemical Cycles, in: Climate Change 2013: The Physical Science Basis. Contribution of Working Group I to the Fifth Assessment Report of the Intergovernmental Panel on Climate Change, edited by Stocker, T., et al., pp. 465–570, Cambridge University Press, Cambridge, UK, and New York, NY, USA, 2013. Friedlingstein, P., Dufresne, J.-L., and Cox, P. M.: How positive is the feedback between climate change and the carbon cycle?, Tellus (2003), 55B, 692–700.

Friedlingstein, P., Cox, P., Betts, R., Bopp, L., von Bloh, W., Brovkin, V., Cadule, P., Doney, S., Eby, M., Fung, I., Bala, G., John, J., Jones, C. D., Joos, F., Kato, T., Kawamiya, M., Knorr, W., Lindsay, K., Metthews, H. D., Raddatz, T., Rayner, P., Reick, C., Roeckner, E., Schnitzler, K.-G., Schnur, R., Strassmann, K., Weaver, A. J., Yoshikawa, C., and Zeng, N.: Climate-Carbon Cycle Feedback Analysis: Results from the C4MIP Model Intercomparison, Journal of Climate, pp. 3337–3353, 2006.

**Point-by-point response to comments by Reviewer 2**

We thank Reviewer 2 for her/his constructive comments. In the following, we respond to her/his comments (cited by using *grey*, *italic font*) and explain how we addressed the points raised in the preparation of the revised version of our manuscript.

Authors compare carbon cycle feedbacks from a pre-industrial and LGM simulation using the framework described by Friedlingstein et al. (2006) and Arora et al. (2013). Overall although the result may be somewhat obvious I still see this as a useful study as long as the underlying mechanisms are thoroughly investigated. However, the manner in which the manuscript is currently written shows that the authors haven't gained a sufficient understanding of the science as well as terminologies used in the existing literature. As such then it is clearly not of publication quality in its current form.

**Main comments**

My biggest concern is with the equations. On page 5 I\_tot is not defined (unless I missed it) but if I try to interpret I\_tot it seems like the change in atmospheric CO2 burden. I\_ext on the other hand is total cumulative emissions. If true, then the ratio between the two (equation 7) is not the feedback but rather the airborne fraction. This is not the way Friedlingstein et al. (2006) or Arora et al. (2013) described the feedback and the gain. Their feedback and the gain are calculated by comparing either simulated CO2 (in emissions-driven simulation) or diagnosed emissions (in concentration-driven simulations) from fully-coupled and biogeochemically-coupled simulations."

In the new manuscript, we made sure that every symbol used in our equations is explicitly introduced in the text adjacent to its first appearance. We emphasise that the fraction of I\_tot and I\_ext has been interpreted as the airborne fraction and total feedback strength by Gregory et al. (2009). Friedlingstein et al. (2006) and Arora et al. (2013) focus on the carbon-climate effect when calculating feedback strength and gain, whereas we follow Gregory et al. (2009) and quantify the combined feedback due to carbon-climate and carbon-concentration effect. Moreover, due to a comment of the editor we realized that we have to include the ocean contributions in this equation to make clear what the terrestrial feedback quantities that we compute mean for the total feedback that als includes ocean contributions. Therefore, in the revised manuscript, we now present the feedback formalism in a hopefully more transparent way.

I am also troubled by the fact that in Figure 7 the rate of carbon uptake by land shows an abrupt slow down around CO2 concentration of 650 ppm. Figure 3c of Arora et al. (2009) shows how photosynthesis changes per unit increase in CO2 based on the standard biochemical equations for photosynthesis. Although this rate decreases, because of the saturating effect, I do not see any abrupt changes up until CO2 of 747 ppm in their figure. This abrupt behaviour in authors' model, it seems, doesn't come from the photosynthesis equations but rather something else that is implemented in the model."

This observation does not represent a contradiction between the two studies: Figure 7 in our paper shows the transition between the two 'modes' of photosynthetic assimilation as a function of CO2 concentration each limiting the assimilation rate in its own way: carboyxlation and electron transport. In order to assess which of the two limitations presents a larger constraint to the assimilation rate our model calculates carboxylation rate and electron transport rate and calculates the resulting assimilation rate based on the smaller of the two. This is the same in the CTEM model

used by Arora et al. (2009) – see their equations (4) and (5) that describe the two modes. The difference between our Figure 7 and Figure 3c in Arora et al. (2009) is thus that we display how the assimilation rate resulting simultaneously from both limitations depends on CO2, whereas Arora et al. 2009 show the CO2 dependence of carboxylation and electron transport rate individually. We included Figure 7 into our manuscript to point out that assimilation rates are considerably less sensitive to rising CO2 concentrations after CO2 has reached the transition point from carboxylation to electron transport rate limitation. With the preindustrial CO2 concentration being closer to the transition point than the glacial CO2 concentration, the concentration where the assimilation rate is mainly limted by electron transport is reached much earlier during the PI experiment. This is the main cause for the smaller final sensitivity to rising CO2 concentrations under pre-industrial conditions.

The lack of understanding of the current literature, or perhaps it's just the first language issue, is seen in several phrases used by the authors which do not appear to make any sense. These include "fertilization and radiation effect to the different vegetation distribution", "sensitivities to the fertilization and radiation effect", "when structural limits are hit", "the point of effectivity change", "physiological limits are hit more frequently", "photosynthesis exploitation of the insolation", and "tropical living conditions deteriorate". "factorial simulations" are referred to as "factor simulations"

We revised our use of language and terminology thoroughly in preparation of the new manuscript, particularly in the phrases mentioned by the reviewer, but also throughout the text.

We thank Reviewer #2 for providing his scan with hand written comments to our paper.

This is a change/reaction to a comment by Reviewer 1
This is a change/reaction to a comment by Reviewer 2
This is a change/reaction to a comment by the Editor
This is a change/reaction to a comment by two or more of the above
Language and writing style has been reviewed. We didn't mark all changes to sentence structure but all of the cases
mentioned by reviewers and more have been addressed. The introduction, the methods chapter, and the discussion
have been completely rewritten, and the two sections where we analyze our simulation data were partly reformulated.
Changes in response to particular comments of one of the reviewers or the editor are highlighted ase indicated above.

5

10

Abstract. In simulations with the MPI Earth System Model we study the feedback between the terrestrial carbon cycle and atmospheric  $CO_2$  concentrations under ice age and interglacial conditions. We find different sensitivities of terrestrial carbon storage to rising  $CO_2$  concentrations in the two settings. This result is obtained by comparing the transient response of the terrestrial carbon cycle to a fast and strong atmospheric  $CO_2$  concentration increase (roughly 900 ppm) in C4MIP type simulations starting from climates representing the last glacial maximum (LGM) and pre-industrial times (PI). In this setup

- we disentangle terrestrial contributions to the feedback from the carbon-concentration effect, acting biogeochemically via enhanced photosynthetic productivity when  $CO_2$  concentrations increase, and the carbon-climate effect, which affects the carbon cycle via greenhouse warming. We find that the carbon-concentration effect is larger under LGM than PI conditions because photosynthetic productivity is more sensitive when starting from the lower, glacial  $CO_2$  concentration and  $CO_2$  fertilization
- 15 saturates later. This leads to a larger productivity increase in the LGM experiment. Concerning the carbon-climate effect, it is the PI experiment in which land carbon responds more sensitively to the warming under rising  $CO_2$  because at the already initially higher temperatures tropical plant productivity deteriorates more strongly and extra-tropical carbon is respired more effectively. Consequently, land carbon losses increase faster in the PI than in the LGM case. Separating the carbon-climate and carbon-concentration effects, we find that they are almost additive for our model set-up, i.e. their synergy is small in the global
- 20 sum of carbon changes. Together, the two effects result in an overall strength of the terrestrial carbon cycle feedback that is

almost twice as large in the LGM experiment as in the PI experiment. For PI, ocean and land contributions to the total feedback are of similar size, while in the LGM case the terrestrial feedback is dominant.

**1 Introduction**

The introduction is completely re-written. In particular we omitted the discussion of the climate sensitivity, which confused all reviewers and is not central for our study anyway.

- 5 At the last glacial maximum (21 000 yrs before present, from now on LGM), global mean surface temperature was 4 to 5°C lower than today (Annan and Hargreaves, 2013). Vegetation was not only less widespread but also primary productivity was smaller (Prentice and Harrison, 2009). This was the consequence of the lower  $CO_2$  concentrations during those times (about 200 ppm less than today), acting physically via the resulting lower temperatures (greenhouse effect), and biogeochemically via the reduced photosynthetic activity due to less available  $CO_2$  in the atmosphere (reduced  $CO_2$  fertilization) (Prentice and
- 10 Harrison, 2009). From measuring isotopic carbon composition in ocean sediment cores (Bird et al., 1996) and the isotopic oxygen composition of air trapped in ice cores (Ciais et al., 2012) it has been estimated that terrestrial carbon storage was several hundred gigatons less than today. This is consistent with less primary productivity whose effect on carbon storage must have been larger than the reduction in soil respiration by the lower temperatures (Prentice and Harrison, 2009). This describes how  $CO_2$  shaped the terrestrial carbon cycle at the LGM. But the terrestral carbon cycle acts also back on the atmospheric  $CO_2$
- 15 concentration. Hence one may wonder whether the strength of this feedback was different from today at glacial times. This is what we investigate in the present paper by performing Earth system simulations for conditions of the last glacial maximum and pre-industrial (PI) times. Indeed one could ask this question also for the oceanic carbon cycle component, but this paper focuses on the terrestrial component, which will be shown to dominate the difference in feedback strength between the two Earth system states.
- To quantify the feedback between carbon cycle and climate, Friedlingstein et al. (2003) introduced two sensitivities characterizing the change in stored carbon (terrestrial and/or oceanic) due to different drivers: due to biogeochemical effects of changed atmospheric CO2 concentration, called the *carbon-concentration effect* measured by the  $\beta$  sensitivity [PgC/ppm], and due to climate change, called the *carbon-climate effect* measured by the  $\gamma$  sensitivity [PgC/K]. For recent climate, these sensitivities have been quantified in numerous Earth system simulations, especially within the international Coupled Climate
- 25 Carbon Cycle Model Intercomparison Project (C4MIP) (see e.g. Friedlingstein et al. (2006); Ciais et al. (2013)). Attempts to quantify carbon cycle sensitivities for perturbations of climates from even earlier times are rare. The few observational studies relate reconstructions of atmospheric CO2 concentrations to reconstructions of temperature (see Friedlingstein (2015) for a review), but the resulting 'observed' sensitivity estimates of atmospheric CO2 concentration to temperature typically involve the combined carbon-concentration and carbon-climate effect and are thus neither measuring  $\beta$  nor  $\gamma$  as defined by Friedlingstein
- 30 et al. (2003). An exception is the study by Frank et al. (2010), who considered temperature and  $CO_2$  reconstructions for the last Millennium before the industrial revolution: Their estimate should be a good proxy for  $\gamma$  since during this period the changes in atmospheric  $CO_2$  concentration have been only a few ppm so that the carbon-concentration effect should be negligible. The

resulting  $\gamma$  sensitivity turns out to vary in time showing values compatible with the low end of the range of values found in the C4MIP studies for recent climate. Obtained from Earth system simulations of the last Millennium, similar values for  $\gamma$  were obtained by Jungclaus et al. (2010). The compatibility of those  $\gamma$  values obtained for the last Millennium with those from the C4MIP for recent climate may not be that surprising since the climates differ only moderately. On the other hand, the C4MIP

- 5 values are obtained from simulations that perturb the PI climate dramatically ( $\approx$  quadrupling of atmospheric CO2 concentration), while those for the last Millenium are obtained from historical climate and CO2 variations (observed Frank et al. (2010) or simulated Jungclaus et al. (2010)) that are rather moderate so that it is unclear what such a comparison of  $\gamma$  values actually means. To assure comparability, in the present study we adopt the C4MIP methodology to determine carbon cycle sensitivities for past *and* recent times.
- 10 While there have been attempts to determine climate sensitivity for various climates of the deep past (see e.g. PALEOSENSE (2012)), similar studies for carbon sensitivities are apparently missing. Nevertheless, for the climate during the LGM studied here, the underlying carbon-concentration and carbon-climate effects have been isolated in simulations to understand their separate importance for shaping the geographical distribution of vegetation as compared to today (e.g. Claussen et al. (2013); Woillez et al. (2011)). While in these studies it was sufficient to simulate time slices for past and recent times, transient
- 15 simulations are needed to determine carbon cycle sensitivities that could be compared to  $C^4MIP$  values. In the present study we employ a fully coupled General Circulation Model including dynamic vegetation for transient simulations starting either from a climate state representing the LGM or from PI conditions and forced by a strong increase in atmospheric CO2. Letting the CO2 act either physically or biogeochemically, we isolate the individual contributions from the carbon-concentration and carbon-climate effects to changes of the terrestrial carbon budgets. Using this C4MIP type experiment design we quantify their
- 20 contribution not only by computing  $\beta$  and  $\gamma$  for land carbon, but also by performing a factor analysis following Stein and Alpert (1993) to investigate in particular the additivity of the two effects which is a pre-condition to obtain from those two sensitivities the feedback strength.

The paper is organized as follows: First we lay out the design of our simulation experiments. Next, in section 3, we describe the mathematical framework used for our factor and feedback analysis. The analysis of the simulation results starts in section
4 with a description of the two initial climate states representing the LGM and PI conditions (1850 AD). This prepares for the analysis of the transient simulation in section 5, that contains the main results of our investigation. By applying the factor and feedback analysis we demonstrate that the intensity of the considered feedback is very different for last glacial maximum and recent climate and identify the underlying mechanisms explaining the observed differences in system behaviour. The paper concludes with a critical discussion of our results.

**30 2 Experiment set up**

To quantify the feedback between the carbon cycle and atmospheric CO2 concentrations we follow the C4MIP experiment design (Ciais et al., 2013, Box 6.4) in the variant of *concentration driven* simulations. We state explicitly that we are working with concentration driven experiments in this study. This means we investigate the reaction in climate and carbon cycle to a prescribed strong rise in atmospheric CO2. More precisely, we perform a set of four simulations (*ctrl*, *clim*, *conc*, *full*), of which the first three are needed to quantify carbon cycle sensitivities by the C4MIP approach, while the fourth simulation is performed here for a factor separation following Stein and Alpert (1993).

5

10

**We stick with the expression 'factor separation' as this is how Stein and Alpert (1993) named their method**

Starting from a control simulation (*ctrl*) performed at constant  $CO_2$  concentration, three transient simulations forced by rising  $CO_2$  concentrations are performed. In the first of those transient simulations (*conc*) only the carbon-concentration effect is active, which means that the rising  $CO_2$  concentration is "seen" only by the photosynthesis code of the model, while the radiation code constantly "sees" the  $CO_2$  value of the control simulation. Conversely, in the second transient simulation (*clim*) only the carbon-climate effect is active, i.e. only the radiation code "sees" the rising  $CO_2$  concentrations but not the photosynthesis model. In the third simulation (*full*) both effects are simultaneously active. These simulations are run once for LGM and once for PI conditions. – In the following, we will use the term 'experiment' to refer to one of the two cases LGM or PI. 'Simulation' will refer to one of the four model runs *ctrl, clim, conc* or *full*.

The CO2 concentrations for the *ctrl* simulations of the two experiments are 185 ppm (LGM) and 285 ppm (PI), which are
also the initial conditions for the respective transient simulations. Experiments were performed with MPI-ESM (see below). In fact, we performed only the LGM experiment for this study since we could use the published MPI-ESM CMIP5 simulations *piControl, esmFdbk1, esmFixClim1* and *1pctCo2* for our purpose that were performed for PI conditions with the same model version.

These are the official simulation names used in the CMIP5 protocol (Taylor et al., 2012).

20 The LGM simulations were initialized from restart files of the MPI-ESM CMIP5 last glacial maximum spin-up experiment (1800 simulation years long), extended by another 200 years with dynamic vegetation now switched on. The PI simulations used for our study were initialized from a spin-up experiment covering more than 3000 years. For the transient simulations *clim, conc* and *full*, the same atmospheric CO2 concentration increase

We state explicitly that the simulated  $CO_2$  concentration increase is the same in both experiments and explain that this is done to avoid ambiguity in the attribution of different simulation results to different initial Earth system states.

- is imposed over a period of 150 years in both experiments (see Fig. 1), acting differently in the three simulations as explained above. The forcing for our LGM experiment is obtained by reducing the standard PI  $CO_2$  forcing by 100 ppm to account for lower glacial  $CO_2$  concentrations while preserving the rate of change. Because  $CO_2$  concentrations thereby increase by the same amount, the different reaction of the Earth system to the  $CO_2$  rise in the two experiments should mostly be attributable to the different initial conditions, i.e. the glacial-interglacial atmospheric  $CO_2$  offset and the particular characteristics of the
- 30 initial climates. The distribution of ice sheets is prescribed to the appropriate LGM and PI conditions and is kept constant in all simulations.

The experiments are conducted with the Earth-System Model of the Max Planck Institute (MPI-ESM) using the version described in Giorgetta (2013). The MPI-ESM consists of the atmosphere component ECHAM6 and the ocean component

**Figure 1.** *CO*2 change scenarios as prescribed for the LGM and PI experiments: Starting from 185ppm ("Last Glacial Maximum", green line) and starting from 285ppm ("Pre-Industrial", red line).

MPIOM, both including submodels for simulating the land and ocean carbon cycles. Because atmospheric  $CO_2$  concentrations are prescribed in our experiments, the oceanic and terrestrial carbon cycles are decoupled so that changes in the ocean carbon cycle are irrelevant for terrestrial carbon reservoirs that are of main interest here; nevertheless oceanic carbon fluxes play a role for calculating the overall carbon cycle feedback in our study and the physical ocean remains an important component

- 5 of the climate dynamics affecting also the land carbon cycle. The land component JSBACH comprises the DYNVEG model for simulation of natural changes in the geographical distribution of vegetation controlled by competition and wind and fire disturbances (Reick et al., 2013), and the BETHY model (Knorr, 2000) for simulation of the fast biochemical and biophysical processes of the biosphere, in particular photosynthetic production that is simulated following the Farquhar model (Farquhar et al., 1980) for C3 and the Collatz model (Collatz et al., 1992) for C4 plants. Vegetation is represented by eight plant functional
- 10 types that differ in phenology and physiology and interact dynamically (see Brovkin et al. (2013) for an evaluation of the present implementation of dynamic biogeography). There is no anthropogenic land cover change considered in the experiments here. Terrestrial carbon dynamics are calculated with CBALANCE (Reick et al., 2010), representing vegetation, litter, and soils by seven carbon pools, where temperature dependence of heterotrophic respiration is modeled by a Q10-formula and turnover rates are in addition dependent on soil humidity. The oceanic biogeochemistry model HAMOCC (Ilyina et al., 2013) calculates
- 15 sea-air gas exchange, water column processes and sediment dynamics.  $CO_2$  exchange between sea and air is calculated with a temperature dependent rate based on the thermodynamic disequilibrium at the interface. In the water column, it is cycled as organically fixed carbon, dissolved inorganic carbon and calcium carbonate and is exchanged with sediments in the latter two forms. Temperature, nutrient and light dependent biological cycling of carbon within the water column is represented by an extended NPZD model (Six and Maier-Reimer, 1996), inorganic carbon cycling is based on Maier-Reimer and Hasselmann
- 20 (1987), using updated chemical constants by Goyet and Poisson (1989).

**3 Analysis framework**

This section has been completely rewritten, because all reviewers criticised our way to derive the terrestrial contribution to the carbon cycle feedback. We now explicitly include the ocean into our considerations and show that feedback factors for ocean and land add to the overall feedback factor (see eq. (11)), so that we can separate the terrestrial contribution without need to do a similar sensitivity analysis for the ocean.

Here we introduce the mathematical framework for analyzing our simulations in the next sections. First we describe how we apply the factor separation method by Stein and Alpert (1993) to separate the relative contributions of the carbon-concentration and carbon-climate effects to the overall changes in terrestrial carbon reservoirs.

We now use the names 'carbon-concentration effect' and 'carbon-climate effect' instead of other formulations that the reviewers found inappropriate.

In the remainder of the section we describe the mathematical framework to disentangle the oceanic and atmospheric contributions to the overall carbon cycle feedback, as well as the contributions of those two effects to the feedback. This feedback framework was originally introduced by Friedlingstein et al. (2003) and further discussed by Gregory et al. (2009). We apply

10 it here in the variant with prescribed atmospheric  $CO_2$  (Ciais et al., 2013, Box 6.4).

We apply the factor separation method of Stein and Alpert (1993) as follows. Let  $C_L$  denote the total land carbon. The pure effects of the carbon-concentration and carbon-climate effects are individually quantified by the differences

$$\Delta C_{L,conc}(t) := C_{L,conc}(t) - \overline{C}_{L,ctrl}$$

$$\Delta C_{L,clim}(t) := C_{L,clim}(t) - \overline{C}_{L,ctrl}$$
(1)

We set a colon before the equal sign to indicate that the term at the side of the colon is defined by the other side. We feel that this helps to make immediately obvious what is a definition, and what is a derived relation.

where the indices at the right hand side  $C_L$ -values refer to the simulations from which the values were obtained, while the indices to the  $\Delta C_L$ -values at the left hand side refer to the effect considered. The time dependence t appears only for the values from transient simulations, but not for values from the control simulations which enter our calculations as mean values (indicated as a bar over the symbol). In addition, we quantify the 'synergy' between the carbon-concentration and the carbon-climate effects, which is that part of the land carbon storage difference between the *full* and *ctrl* simulation that cannot

20

15

5

be explained by a linear addition of the individual effects:

We stick with the term 'synergy' because this is the name for the non-linear contributions introduced by Stein and Alpert (1993) and subsequently used in studies applying this method.

$$\Delta C_{L,syn}(t) := (C_{L,full}(t) - \overline{C}_{L,ctrl}) - (\Delta C_{L,conc}(t) + \Delta C_{L,clim}(t)).$$
(2)

Note that in this way all separate factors sum up to the land carbon change in the *full* simulation:

25
$$\Delta C_{L,full}(t) = \Delta C_{L,conc}(t) + \Delta C_{L,clim}(t) + \Delta C_{L,syn}(t).$$
(3)

For the feedback analysis we consider the following differences in near surface temperature and atmospheric  $CO_2$  concentration that develop in the transient simulations:

$$\Delta T_{clim}(t) := T_{clim}(t) - \overline{T}_{ctrl}$$

$$\Delta cc(t) := cc(t) - cc_{ctrl}.$$
(4)

We changed our symbol for the concentration of atmospheric  $CO_2$  from *c* to *cc* to avoid confusion with our symbol for carbon storage *C*.

[revised manuscript text omitted]

- 10

We corrected 'indonesian archipel' to 'Malay Archipelago' because the former was not correct in this context.

On global scale, less area is covered by vegetation and dense vegetation is restricted to the tropical zone (compare Fig. 3). In the PI state, vegetation reaches far more into the extratropics and the mid latitudes are more densely covered by vegetation. Terrestrial carbon reservoirs are larger in the PI experiment almost everywhere (see Fig. 3). Globally, terrestrial carbon reservoirs contain 1986 PgC in the LGM and 3041 PgC in the PI state. Our difference in carbon storage (1055 PgC) matches the

15 difference of  $1030\pm625$  PgC in non-permafrost land carbon obtained by Ciais et al. (2012) from combining model simulations with carbon and oxygen isotope data from sediment and ice cores; note that changes in permafrost carbon are not part of our simulations.

We included now a map of initial terrestrial carbon reservoir sizes and report also global totals.

(b) PI - LGM plant water availability

(a) PI -LGM yearly mean 2m air temperature

**Figure 2.** Differences between the LGM and PI climates obtained in the respective *ctrl* simulations: a) Difference in global mean near surface temperatures and b) difference in plant water availability. Here the values in the LGM state are substracted from the values in the PI state. Land areas that are covered by ice in the LGM but not in the PI equilibrium state show soil humidity differences > 0.4.

---

## Referee Report (RR1)

This paper uses an ESM to study the different interactions of the physical and biogeochemical aspects of climate at the LGM and PI. It uses these two cases to make some more general comments on this interaction and how it is treated. I note that I was asked to review only the revised version of the manuscript. I did not read the initial version although I did read enough of the responses to see that the authors have largely rewritten it and that the initial critiques were robust. My comments are based only on the revised version.

My first thought when reading the paper and responses is that the terminology and even mathematics around this field is a hideous mess. The same terms seem to be used to describe sensitivity of fluxes to perturbed forcings or the sensitivity of equilibrium stocks. The same term is also used to describe the total role of the carbon cycle in mediating the relationship between anthropogenic emissions and climate change or the modulation of this role by climate impacts on the carbon cycle. Here we also see sensitivities of transient changes in stocks. the authors can't avoid getting entangled in this mess and, as reviewers, we can't blame them for it. the authors can, though, avoid confusing the issue further and getting caught up in interpretation of quantities that probably depend on details of their simulation.

I also sympathise with the authors' dilemma. They have a relatively clear (perhaps obvious) model result. they properly try to analyse and generalise this result using diagnostics developed for other reasons. Much of the controversy relates to this analysis rather than the result itself. Previous reviewers were obviously concerned about aspects of this analysis and I will add to those concerns. The nature of the critiques does, though, suggest a way out. I recommend the authors focus their analysis on the model results themselves. Where are the differences between LGM and PI semsitivities greatest, what processes contribute to these differences. There is much of this material already in the paper and probably more that could be taken from the simulations. I am not sure that what the authors have to say about $\beta$ and $\gamma$ rewards the difficulties it has caused them. I also think there are difficulties with this analysis. For example, quoting the time-dependent changes in sensitivity seems quite risky, since it may well arise from an interaction of the time-scales of the transient forcing convolved with climate and carbon-cycle time-scales. We can't easily tell and it would seem to be a lot of work to disentangle the effects.

I believe there is a useful paper within the material the authors present. I believe this will be a shorter, more focused paper. I hope the authors will persevere and revise the manuscript.

---

## Editor Decision (ED1)

Dear Dr. Adloff,

Thank you for submitting your manuscript to Earth System Dynamics and for providing responses to the points which the two referees raise. On the basis of the critics by the referees alone, one may (have to) decide to reject the paper in its current form for publication. The use of the terms "sensitivity", "feedback", "gain", and "feedback factor" would need improved clarification. It is a good idea to follow the corresponding terminology of the IPCC where possible and to describe where deviations from this terminology are made. Some of the confusion unfortunately is intrinsic to the climate feedback discussion, because in Hansen et al. (1984) and many subsequent papers of other authors the terms "gain" and "feedback factor" are used in the opposite way as originally phrased in electrical engineering. See Roe (2009) for a sound discussion of the feedback terminology. I discussed the manuscript with two further experts in the field to arrive at a conclusion. There independent response was more positive than that of the two reviewers. The authors make constructive suggestions for improving the manuscript. My own impression is that the paper addresses a basic simple question, which should be clarified at some point. I think the underlying idea is a good one and a novel one. Weighing all factors, I would like to give the authors a chance to make a major revision of their manuscript. This decision I do with some reservations, because only a substantially improved manuscript would have a chance to pass a new round of reviews.

Some further recommendations resulting from discussing this manuscript:

The use of terminology is partly wrong ("climate sensitivity" as pointed out by the reviewers), or inconsistent with existing literature ("fertilisation" and "radiation" effects are usually referred to as carbon-concentration and carbon-climate feedbacks). It would be better if the authors stick to these well-established terms.

Importantly, the equation (6), and consequently (7) and (8) cannot be used here, because this would require the ocean feedbacks (also pointed out by #2). Therefore, Fig. 8 also does not make sense. One could argue that I_ext in (6) was the compatible emissions if the ocean sink was absent, but I am not sure whether this would make sense. In any case this would need a more detailed explanation and justification.

Page 4, line 14-15, and later page 7, line 13: The finding that (Delta C_fert + Delta C_rad) = Delta C_full is a very model specific feature. Both, larger and smaller sums have been found (see e.g. Gregory et al. 2009, cited).

The fact that Delta T_rad < Delta T_full is a well known feature (see also Gregory et al. 2009 and references therein), and this needs to be mentioned.

If you decide to submit a revised version of your manuscript, please, take all points of the referees and these mentioned here into account. I would then send out this manuscript for a new review round. Thank you for using the open access way for publishing your work.

Best regards, Christoph Heinze

References:

Hansen, J., A. Lacis, D. Rind, G. Russell, P. Stone, I. Fung, R. Ruedy, and J. Lerner (1984), Climate Sensitivity: Analysis of Feedback Mechanisms, in Climate Processes and Climate Sensitivity, edited by J. a. T. Hansen, T., pp. 130-163, American Geophysical Union, Washington, DC.
Roe, G. (2009), Feedbacks, Timescales, and Seeing Red, Annu Rev Earth Pl Sc, 37, 93-115, doi: 10.1146/annurev.earth.061008.134734.

---

## Author Response (AR2)

Dear Dr. Heinze,

Thank you for the encouraging words! Based on your and the reviewer's comments, we revised our manuscript and changed some paragraphs, as highlighted in the annotated manuscript version.

Anonymous Reviewer 3 pointed out additional literature that we could refer to and some word choice issues. We addressed all of these as indicated in our annotated manuscript and in the response to Anonymous Reviewer 3.

The concerns of Peter Rayner are twofold: He is critical about the usage of the $\alpha - \beta - \gamma$ framework and encourages us to reconsider the discussion of the time dependence of the sensitivity parameters. As written in our response to him, we feel that using a different metric for our study would require to redo the study as a whole. We do not think that this would be necessary, given that their usage allows us to identify key differences in a glacial and interglacial Earth system state that determine the respective carbon cycle sensitivities to increasing $CO_2$ concentrations. Peter Rayner's comment on the scenario and time dependencies of the sensitivity parameters led us to extend the discussion of our results and state more clearly that we carefully chose our experiment design in a way that the same forcing is applied to two different Earth system states. When comparing sensitivity parameters obtained this way, we are confident to have minimized effects of the scenario on our results.

We have increased the legibility of the scales on our figures and hope that the new version of the manuscript is acceptable for publication.

Best regards,
Markus Adloff, Christian Reick and Martin Claussen

Point-by-point response to comments by Reviewer 3

We thank the Anonymous Reviewer 3 for her/his constructive comments on our manuscript. In the following, we address each of the comments individually.

*Reviewer:*
*I think the authors have done an excellent job in revising their manuscript. All concerns of the reviewers have been adequately addressed. I would recommend this manuscript for publication after a few rather minor remarks and corrections have been taken into consideration.*
*(please note page and line numbers refer to the annotated version of the revised manuscript)*

Our response:
We are pleased to read that we have met the reviewer's expectations.

*Reviewer:*
*page 3, lines 2: Check sentence (delete on "obtained"); consider replacing one of the two "obtained" in the two following sentences.*

Our response:
The reviewer spotted one obsolete word which we deleted. We also followed the suggestion to change the repetitive usage of 'obtained' in the following sentences.

*Reviewer:*
*page 4, lines 2-4: I find the perspective that the "full" simulation is needed for a factor separation a bit weird. There is no reason to expect that the "clim" and "conc" simulations would add up, and it has been shown that they in general do not (Gregory et al. 2009, Zickfeld et al 2010, Schwinger et al. 2014). Therefore, the "full" simulation is needed to get the correct feedback (in the model world), the "clim" simulation can be seen as necessary for a factor separation.*

Our response:
We fully agree and thank the reviewer for the additional literature hints. In the hope to prevent the irritations expressed in the reviewer's comment, we reformulated the respective sentences as:
"To quantify the feedback between carbon cycle and atmospheric $CO_2$ concentrations we combine the $C^4MIP$ experiment design (Ciais et al., 2013, Box 6.4) in the variant of concentration driven simulations with a factor separation following Stein and Alpert (1993). Technically, we proceed by investigating the reaction in climate and carbon cycle to a prescribed strong rise in atmospheric $CO_2$. More precisely, we perform a set of four simulations called "ctrl", "clim", "conc", and "full". While for the quantification of the feedbacks by the $C^4MIP$ approach only three of these simulations are needed, by using the full set of all four simulations we are able to demonstrate that -- in contrast to other models (Gregory et al. 2009, Zickfeld et al. 2010, Schwinger et al. 2014) -- the linearity assumption implicit to the $C^4MIP$ feedback analysis is indeed justified for our model."

*Reviewer:*
*page 4, line 17: Is it necessary to spell out the CMIP5 simulation names (piControl, esmFdbk1,...) here? This is rather technical, and the abbreviations are not used in the following text.*

Our response: We provide the technical names of the simulations to give a clear reference the previously published simulations we are using. However, we state them in brackets to indicate them as addition of low importance.

*Reviewer:*
*page 5, line 17: In HAMOCC, organic carbon is sedimented, so it is not only "the latter two forms"*

Our response:
We thank the reviewer for pointing out this misleading formulation and corrected it.

*Reviewer:*
*page 7, line 7: experment -> experiment*

Our response:
We corrected this spelling mistake.

*Reviewer:*
*page 8, lines 28-29: check use of parentheses*

Our response:  We removed a wrongly placed parenthesis

*Reviewer:*
*page 10, line 8-9: It might be good to mention that less evapotranspiration is caused by increased stomatal closure. That this leads to a radiative forcing via reduction in low level clouds has been shown e.g. by Doutriaux-Boucher et al. (doi:10.1029/2008GL036273, GRL 2009)*

Our response:
We thank the reviewer for her/his suggestion and added the proposed reference to our text.

*Reviewer:*
*page 11, line 4: Please check sentence ("due to climate change dominates" sounds odd).*

Our response:
We changed the respective sentence and hope the new formulation is clearer.

Point-by-point response to comments by Reviewer 4

We thank the reviewer, Peter Rayner, very much for his constructive comments and suggestions. In the following, we respond to the reviewer's comment.

*Reviewer:*
*This paper uses an ESM to study the different interactions of the physical and biogeochemical aspects of climate at the LGM and PI. It uses these two cases to make some more general comments on this interaction and how it is treated. I note that I was asked to review only the revised version of the manuscript. I did not read the initial version although I did read enough of the responses to see that the authors have largely rewritten it and that the initial critiques were robust. My comments are based only on the revised version.*

*My first thought when reading the paper and responses is that the terminology and even mathematics around this field is a hideous mess. The same terms seem to be used to describe sensitivity of fluxes to perturbed forcings or the sensitivity of equilibrium stocks. The same term is also used to describe the total role of the carbon cycle in mediating the relationship between anthropogenic emissions and climate change or the modulation of this role by climate impacts on the carbon cycle.*

Our reply:
Peter Rayner strengthens the impression which we also got from reviewers in the first round of reviews that terminology is easily misleading in this area of research and key to avoiding confusion. It was indeed our main concern in reaction to those reviewer's comments to make the mathematical feedback framework that we use for our analysis as transparent as possible.

*Reviewer:*
*Here we also see sensitivities of transient changes in stocks. the authors can't avoid getting entangled in this mess and, as reviewers, we can't blame them for it. the authors can, though, avoid confusing the issue further and getting caught up in interpretation of quantities that probably depend on details of their simulation. I also sympathise with the authors' dilemma. They have a relatively clear (perhaps obvious) model result. they properly try to analyse and generalise this result using diagnostics developed for other reasons. Much of the controversy relates to this analysis rather than the result itself. Previous reviewers were obviously concerned about aspects of this analysis and I will add to those concerns. The nature of the critiques does, though, suggest a way out. I recommend the authors focus their analysis on the model results themselves. Where are the differences between LGM and PI sensitivities greatest, what processes contribute to these differences. There is much of this material already in the paper and probably more that could be taken from the simulations. I am not sure that what the authors have to say about β and γ rewards the difficulties it has caused them.*

Our reply:
Probably Peter Rayner's suggestion arises from his – understandable – conceptual concerns with the whole $\alpha - \beta - \gamma$ feedback framework that we use for our analysis. We still think that the meaning of the $\alpha - \beta - \gamma$ sensitivities is sufficiently obvious to provide useful measures to understand why in our study differences arise in the response of the carbon cycle to the same relative increase in atmospheric $CO_2$ starting from preindustrial climate and LGM climate, respectively. In particular, our whole experiment design is tailored for this type of analysis related to carbon-concentration and carbon-climate feedback. Accordingly, we think that following Peter Rayner's suggestion not to use the $\alpha - \beta - \gamma$ sensitivities, our study would become a completely new and completely different study. Hence we prefer to not consider his suggestion.

*Reviewer:*

*I also think there are difficulties with this analysis. For example, quoting the time-dependent changes in sensitivity seems quite risky, since it may well arise from an interaction of the time-scales of the transient forcing convolved with climate and carbon-cycle time-scales. We can't easily tell and it would seem to be a lot of work to disentangle the effects. I believe there is a useful paper within the material the authors present. I believe this will be a shorter, more focused paper. I hope the authors will persevere and revise the manuscript.*

*Our reply:*

We fully agree that the time dependence of the sensitivities arises from "*the time-scales of the transient forcing convolved with climate and carbon-cycle time-scales*". And this, indeed, runs contrary to the idea that a 'sensitivity' should characterize a system as such without being affected by the way it is forced. This circumstance is well known (e.g. Gregory et al. 2009, Arora et al. 2013). Nevertheless, as we show in our study, by discussing the full time dependence of the sensitivities we gain considerable insight into system behaviour - by investigating, for example, the non-linear change of $\beta$ (Fig. 6) and linking it to thresholds in the productivity scheme. Resorting to quoting $\alpha - \beta - \gamma$ only for a particular time horizon (as in Table 2) would have deprived us of this opportunity. Moreover, we would then have to defend our particular choice of the time horizon. This is impossible, since the choice is intrinsically arbitrary. In addition, our study would then be incomplete insofar as it would not be clear to what extent our results would be independent of the particular time horizon chosen. Therefore, we still think that considering the whole time dependence in our study is sufficiently justified by the additional insight we gain. – When resubmitting our paper, we happily take up the reviewer's point and shortly discuss that indeed the time scales of forcing, climate and carbon combine to the time dependence of the sensitivities. In addition, this gives us the opportunity to stress more clearly that by using similar forcing scenarios of absolute $CO_2$ increase, the time scale dependence of the forcing largely drops out in the comparison of our LGM and PI experiments – we hope that the reviewer agrees that this partially alleviates the problem in using time dependent sensitivities.

References:

[revised manuscript text omitted]

> We clarified that only three out of the four simulations performed are needed for the feedback analysis. The fourth simulation was added to the set to test the assumed linearity 
[revised manuscript text omitted]

---

## Author Response (AR3)

Dear Dr. Heinze,

Thank you for your decission to accept our manuscript for publication, as well as your encouragments and efforts throughout the review process! The critical and constructive discussion of the manuscript helped to improve our text significantly. We considered the suggested editorial changes for the preparation of the final version of the manuscript.

Best regards,
Markus Adloff, Christian Reick and Martin Claussen